# Residual Axial Behavior of Restrained Reinforced Concrete Columns Damaged by a Standard Fire

**Mohamed Monir A. Alhadid and Maged A. Youssef \***

Department of Civil and Environmental Engineering, Western University, London, ON N6A 5B9, Canada; majjanal@uwo.ca
* Correspondence: youssef@uwo.ca

**Abstract:** A simplified procedure to predict the residual axial capacity and stiffness of both rectangular and circular reinforced concrete (RC) columns after exposure to a standard fire provides the means to replace the current descriptive methods. The availability of such a procedure during the design phase provides engineers with the flexibility to come up with better designs that ensure safety. In this paper, finite difference heat transfer and sectional analysis models are combined to determine the axial behavior of RC columns with various end-restraint conditions at different standard fire durations. The influence of cooling phase on temperature distribution and residual mechanical properties is considered in the analysis. The ability of the model to predict the axial behavior of the damaged columns is validated in view of related experimental studies and shown to be in very good agreement. A parametric study is then conducted to assess the axial performance of fire-damaged RC columns. A procedure is proposed to determine the residual strength and stiffness of fire-damaged RC columns in typical frame structures.

**Keywords:** reinforced concrete; columns; standard fire; cooling phase; axial capacity; temperature-stress history





## 1. Introduction

Reinforced concrete (RC) structures are widely used in construction due to their outstanding structural performance and design flexibility [1]. The behavior of RC members at ambient conditions is addressed by various building codes and standards [2–4]. However, when exposed to elevated temperatures, the capacity and deformation of such members change due to material degradation, residual strains, and stress redistribution [1,5,6]. In addition, the temperature–load history and the interaction between mechanical and thermal stresses significantly affect the residual properties of the members [1,7,8].

The structural integrity and mechanical properties of most fire-exposed concrete members are either fully or partially restored after the fire incident. Many design codes and standards [9–12] adopt a prescriptive approach through providing data related to the anticipated fire resistance of various RC members based on their geometrical properties and fire exposure conditions. This approach is easy to implement but usually results in bigger sections than what is required to support the loads. The prescriptive approach also overlooks the influence of temperature–load history despite its important role in determining the residual performance of the members. In practice, a preliminary assessment of the damaged members is performed immediately after the structure is exposed to elevated temperatures [13]. This includes visual inspection, hammer tapping, determination of fire propagation route and residual strength of concrete, cracking and spalling schemes, color changes, and smoke deposits to identify the fire duration and maximum temperatures reached [14]. After that, the structure is evaluated according to the relevant design code based on the extent of damage and the affordability of the required work. Load-bearing members, such as columns, should maintain their structural integrity to sustain the applied load without failure or excessive deflections.

This study is an attempt to propose an analytical procedure to supplement the in situ preliminary assessment after a fire incident in RC structures considering standard fire. A model utilizing both heat transfer analysis and sectional analysis is developed to evaluate the residual axial behavior of rectangular and circular RC columns. Temperature–load history is explicitly considered in the analysis. The various strain components developed during and after fire are calculated and their influence on changing the residual performance of the damaged members under various restraining conditions is evaluated. The validity of the proposed model is assessed in view of relevant experimental results obtained from the literature. The validated model is then utilized to perform a parametric study aiming at investigating the influence of mechanical properties, cross-sectional dimensions, fire exposure and support conditions on the residual performance of RC columns. A simplified procedure is then proposed to predict the residual axial capacity and stiffness of RC columns in typical frame structures. The outcomes of the current study provide a solid basis for a more comprehensive work that accounts for other fire types and exposure conditions.

The determination of the residual axial capacity of RC columns subjected to elevated temperatures is not practical in design offices due to the complexity associated with performing comprehensive thermal and structural analyses. The proposed simplified method allows engineers to utilize the commercially available structural analysis software to predict the residual axial capacity and deformation behavior of fire-exposed structures. This can be performed by considering the residual axial stiffness as an input in the definition of the mechanical properties of the affected members to account for the deterioration they exhibited due to elevated temperatures. The internal forces in all the members can then be obtained due to the load redistribution triggered by the fire scenario. The capability of the structural members to resist the applied loads is determined in view of their residual capacity. The residual axial properties used in performing the structural analysis study are obtained from the proposed models in the current article.

## 2. Proposed Analytical Approach

Assessment of the post-fire behavior of RC columns in typical frame structures requires the consideration of not only the residual mechanical properties of the composing materials but also the temperature–load interaction before and during fire. Figure 1 illustrates the influence of heating and loading history on the total strains ($\varepsilon_t$) induced in concrete. Path 1 shows the case where the column supports a load that causes a mechanical strain ($\varepsilon_m)_1$ before heat exposure. By heating the column, a combination of thermal and transient strains ($\varepsilon_{th})_1$ is induced. On the other hand, path 2 shows the development of total strains under a successive application of temperature and load. In this case, the column experiences thermal strains ($\varepsilon_{th})_2$ followed by mechanical strains ($\varepsilon_m)_2$ due to the loads applied on the fire-damaged member. Transient strains are not considered as the column is unloaded during heating. Although the column is supporting the same load and is exposed to the same maximum temperature in both cases, the total strain differs significantly. In real structures, the total strain can be somewhere in between the two previously mentioned extreme cases. Since the free thermal strain is partially irrecoverable and the transient strain is irreversible [7,15], detailed examination of the actual load–temperature path must be considered in the analysis. Guo and Shi [1] experimentally demonstrated the variation in the deformation behaviors of RC columns when subjected to different heating–loading paths.

The analytical approach performed in this study encompasses three main stages that describe the structural variations in the exposed member throughout the heating–cooling cycle. Firstly, the structural performance of the intact member is determined in terms of its capacity and stiffness considering the relevant material models at ambient conditions. The obtained structural characteristics act as a basis to calculate the initial axial load level ($\lambda$) and to determine the extent of deterioration in the member after exposure to fire. The second stage involves thermal and structural analyses of the exposed member during the heating and cooling cycles. Heat transfer analysis is carried out using the finite difference

method to determine the maximum temperature distribution within the member based on concrete thermal and physical properties. In Figure 2, the residual properties of the member at the final stage (point 2) are highly dependent on the temperature–load path followed. Therefore, at each time increment, the change in the applied load level ($\Delta\sigma$) associated with the restraint conditions is considered. Both thermal and transient strains are calculated at each time increment, as represented by the step function shown in Figure 2. The residual capacity of the member during fire is calculated based on the relevant material models to check if failure occurs during fire. The third analysis stage commences after the member is completely cooled down to room temperature. In this stage, sectional analysis is carried out to determine the residual capacity and stiffness of the fire-damaged member in view of the maximum temperature reached and residual strain distribution. The analysis is performed by applying uniform strain increments until failure occurs considering the post-fire mechanical properties and material models.

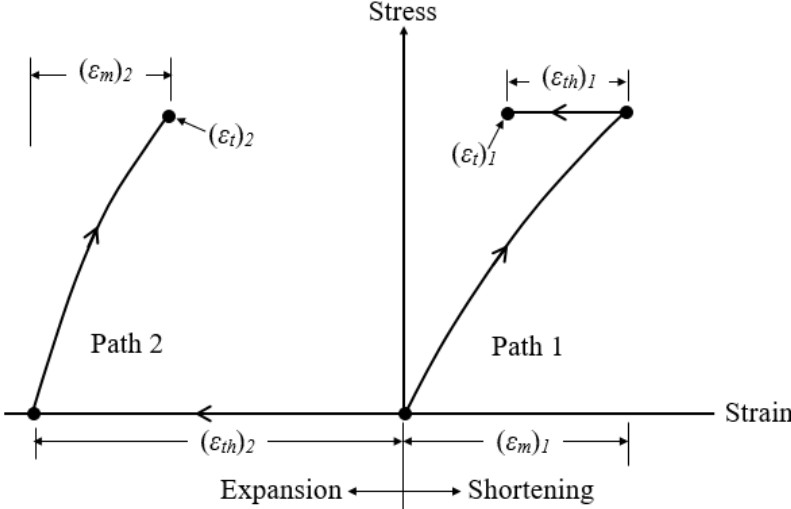

**Figure 1.** Influence of temperature–stress interaction on the concrete strains.

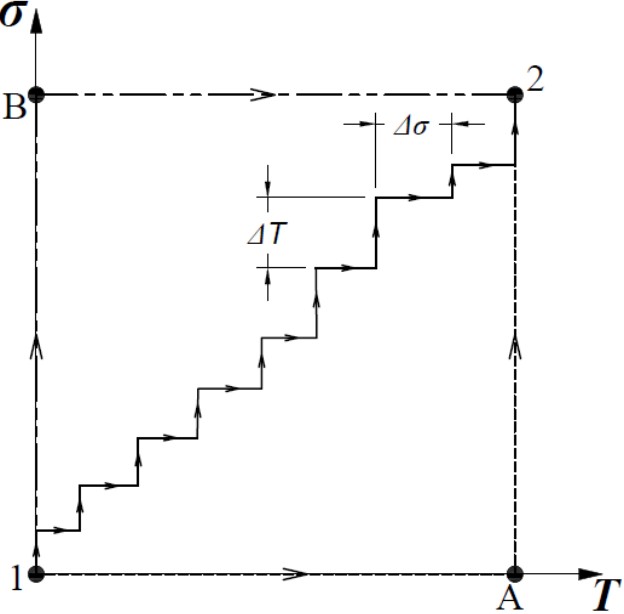

**Figure 2.** Potential temperature–stress paths between initial and final conditions of the fire-exposed members.

The current study focuses on the axial behavior of rectangular and circular RC members exposed to fire from all sides. The restraint condition is determined by performing structural analysis of the entire frame, Figure 3a, with the aid of a suitable commercially available software. The first iteration is performed considering the mechanical properties of the section at ambient conditions. At any specific time during fire, the columns' axial capacity and stiffness are reduced as a function of the temperature distribution within the section. The fire-exposed column can be isolated as shown in Figure 3b. A pin support is assigned to one end of the column, while the other end is attached to a roller support and a spring with an axial stiffness ($k_\delta$) that represents the axial constraints provided by the adjacent frame members. The value of $k_\delta$ can be obtained based on structural analysis, as will be discussed later in Section 11 of this paper. Springs act in resisting the columns' expansion but not contraction. When the column expands, the magnitude of the axial load acting on the column encompasses both the initial applied load ($P_i$) and the restraining force caused by thermal expansion. The axial stiffness ($EA$) of the columns varies at each time step during fire and is considered in the calculation of the restraining force. The mutual dependency is considered in the proposed model, as discussed in the subsequent sections.

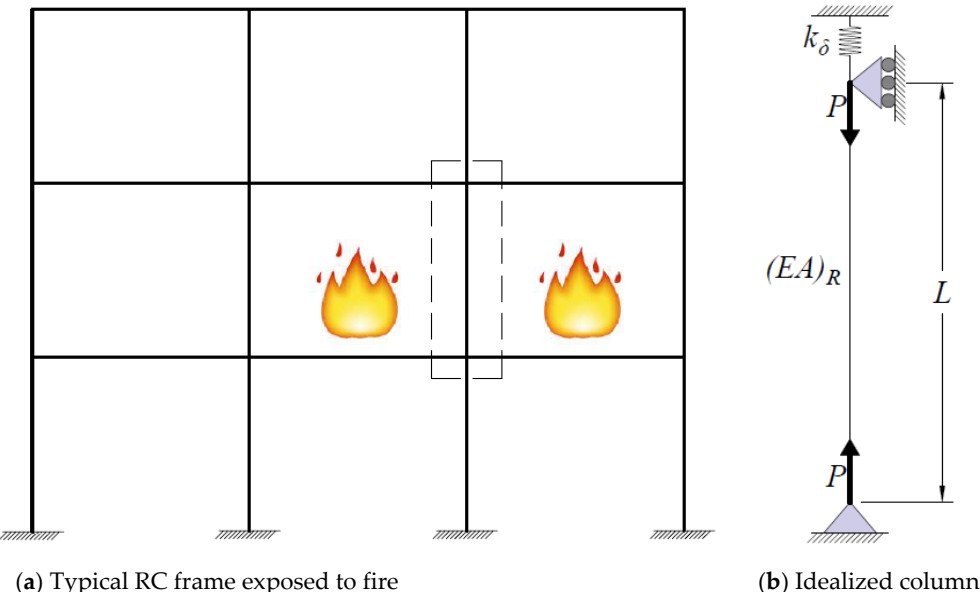

(**a**) Typical RC frame exposed to fire        (**b**) Idealized column

**Figure 3.** Isolation of a fire-exposed column in a typical RC frame.

The proposed analysis of the fire-damaged RC members is carried out based on the following assumptions:

1. Cross-sections remain plane before and after fire. The validity of this assumption was validated for temperatures up to 1200 °C [6].
2. Perfect bond exists between the steel reinforcement and the surrounding concrete.
3. Spalling of concrete is not considered as the analysis is limited to normal-weight concrete.
4. Two-dimensional heat transfer analysis is considered. Thus, heat flow is uniform along the member length.
5. Geometrical nonlinearity is not considered in the analysis.
6. Failure of the compression members is not governed by buckling.

### 3. Definition of Cross-Sections

The residual axial capacity and stiffness of fire-exposed RC rectangular and circular columns subjected to standard fire from all sides are considered in the analysis. The geometrical properties and reinforcement distribution of a typical cross-section are defined in Figures 4a and 5a for rectangular and circular sections, respectively. Rectangular sections are defined in terms of section width ($b$), section height ($h$), steel reinforcement ratio ($\rho$), top

steel reinforcement ($A_{st}$) and bottom steel reinforcement ($A_{sb}$), whereas circular columns are defined in terms of cross-sectional diameter ($D$), steel reinforcement ratio ($\rho$) where steel reinforcement ($A_s$) is assumed to be uniformly distributed along the circumference. Table 1 details the mechanical and geometrical properties of the selected rectangular and circular sections discussed in this paper.

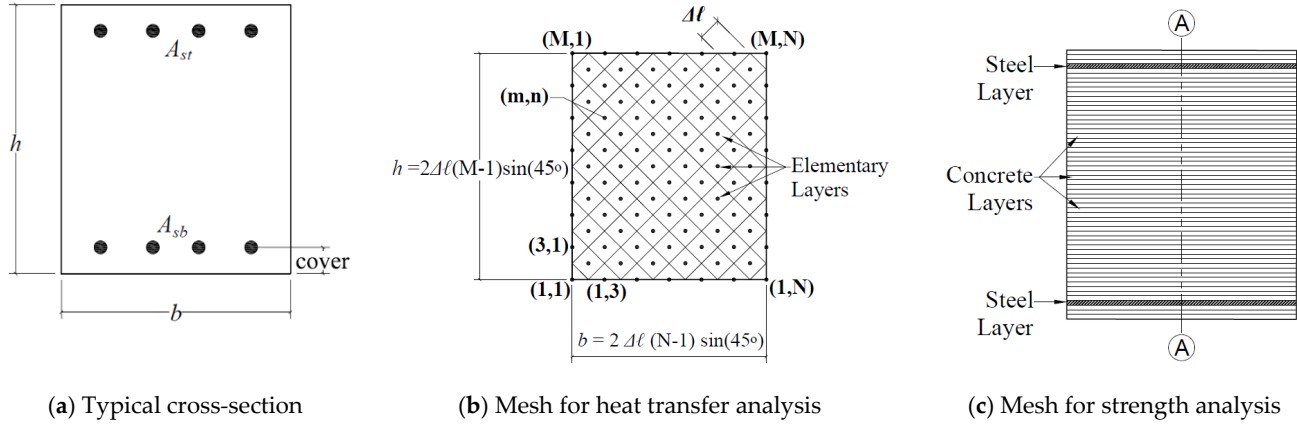

(**a**) Typical cross-section      (**b**) Mesh for heat transfer analysis      (**c**) Mesh for strength analysis

**Figure 4.** Geometry and meshing of rectangular sections.

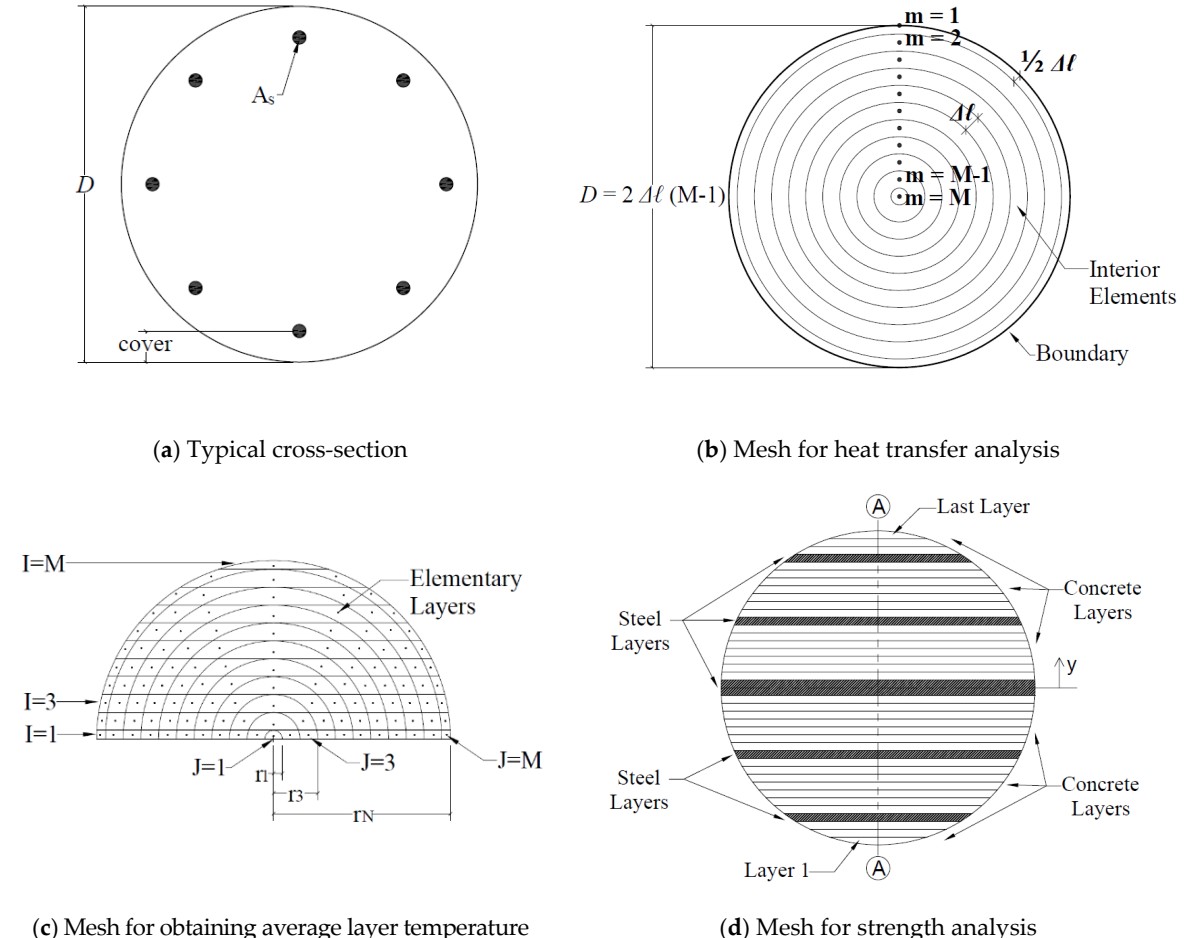

(**a**) Typical cross-section          (**b**) Mesh for heat transfer analysis

(**c**) Mesh for obtaining average layer temperature      (**d**) Mesh for strength analysis

**Figure 5.** Geometry and meshing of circular sections.

**Table 1.** Properties of the considered rectangular and circular column sections.

| Rectangular Sections | | | | | | | | Circular Sections | | | | | |
|---|---|---|---|---|---|---|---|---|---|---|---|---|---|
| Case | $t$ (hr) | $f_c'$ (MPa) | $f_y$ (Mpa) | $b$ (mm) | $h$ (mm) | $\rho$ | $R_D$ | Case | $t$ (hr) | $f_c'$ (MPa) | $f_y$ (Mpa) | $D$ (mm) | $\rho$ | $R_D$ |
| R1 | 1.5 | 35 | 400 | 400 | 500 | 0.04 | 0.0 | C1 | 1.5 | 35 | 400 | 500 | 0.04 | 0.0 |
| R2 | 0.5 | 35 | 400 | 400 | 500 | 0.04 | 0.0 | C2 | 0.5 | 35 | 400 | 500 | 0.04 | 0.0 |
| R3 | 2.5 | 35 | 400 | 400 | 500 | 0.04 | 0.0 | C3 | 2.5 | 35 | 400 | 500 | 0.04 | 0.0 |
| R4 | 1.5 | 25 | 400 | 400 | 500 | 0.04 | 0.0 | C4 | 1.5 | 25 | 400 | 500 | 0.04 | 0.0 |
| R5 | 1.5 | 35 | 300 | 400 | 500 | 0.04 | 0.0 | C5 | 1.5 | 35 | 300 | 500 | 0.04 | 0.0 |
| R6 | 1.5 | 35 | 400 | 250 | 500 | 0.04 | 0.0 | C6 | 1.5 | 35 | 400 | 310 | 0.04 | 0.0 |
| R7 | 1.5 | 35 | 400 | 600 | 500 | 0.04 | 0.0 | C7 | 1.5 | 35 | 400 | 400 | 0.04 | 0.0 |
| R8 | 1.5 | 35 | 400 | 400 | 300 | 0.04 | 0.0 | C8 | 1.5 | 35 | 400 | 780 | 0.04 | 0.0 |
| R9 | 1.5 | 35 | 400 | 400 | 800 | 0.04 | 0.0 | C9 | 1.5 | 35 | 400 | 500 | 0.02 | 0.0 |
| R10 | 1.5 | 35 | 400 | 400 | 500 | 0.02 | 0.0 | C10 | 1.5 | 35 | 400 | 500 | 0.04 | 0.5 |
| R11 | 1.5 | 35 | 400 | 400 | 500 | 0.04 | 0.5 | C11 | 1.5 | 35 | 400 | 500 | 0.04 | 1.0 |
| R12 | 1.5 | 35 | 400 | 400 | 500 | 0.04 | 1.0 | | | | | | | |

## 4. Thermal Analysis

Temperature distribution at any section along the member is determined based on the finite difference method described by Lie [16]. The physical and thermal properties of both concrete and steel are provided by Lie [16]. For each time increment, the temperature distribution within the section is obtained by solving the heat balance equations [16]. In the current study, the columns are exposed to an ASTM E119 [17] standard fire along their perimeter during the heating phase, as approximated by Equation (1).

$$T_f - T_o = 750 \left[1 - e^{(-3.79553 \sqrt{t})}\right] + 170.41\sqrt{t} \tag{1}$$

where $T_f$ is the fire temperature (°C), $T_o$ is the room temperature (°C) and $t$ is the time after the start of the fire (hr). During the cooling phase, the rate of decrease in temperature per minute is calculated according to the ISO 834 [18] specifications as provided in Equation (2) in terms of fire duration at the end of the heating phase ($t_{hot}$).

$$\Delta T = \begin{cases} -10.417 & , & t < 30 \text{ min} \\ -4.167(3 - \frac{t_{hot}}{60}) & , & 30 \text{ min} \leq t < 120 \text{ min} \\ -4.167 & , & t \geq 120 \text{ min} \end{cases} \tag{2}$$

Concrete thermal properties are assumed to be irreversible and maintain a constant value corresponding to the maximum temperature reached [1,15]. A distinction in the meshing procedure between rectangular and circular column sections is illustrated in Figures 4b and 5b, respectively.

### 4.1. Rectangular Sections

The analysis procedure begins by dividing the cross section into M × N 45° inclined square elements, as shown in Figure 4b. The point at the center of each internal element or on the hypotenuse of each boundary element represents the temperature of the entire element. Steel bars are considered as perfect conductors due to their high thermal conductivity, and their temperature is assumed to be identical to the adjacent concrete elements. Heat energy is transferred from the outer elements toward the concrete core, causing a subsequent increase in temperature depending on concrete thermal conductivity and moisture content. The influence of moisture is considered by assuming that when an element reaches a temperature of 100 °C, all the transferred heat causes the evaporation

of water particles instead of raising the element's temperature. Heat transfer equations between the elements throughout the cross-section are given by Lie [16].

Having determined the temperature distribution within the cross-section, the section is divided into multiple horizontal layers, each having a thickness of $\Delta\ell \sin(45°)$, as shown in Figure 4c. Average temperature is then calculated in each layer considering two methods that result in different temperature distribution along the cross-section. In the first one, the temperature of each horizontal layer is calculated as the algebraic average temperature of the square elements composing it. The other calculation procedure is performed by first calculating the residual compressive strength of each square element, and then evaluating the temperature, which would result in the same average compressive strength in that layer. The first temperature distribution is utilized to calculate thermal and transient strains, whereas the second one is used in calculating the residual strength of each layer. The temperature of the steel layer is assumed to be similar to the temperature of the square mesh elements within which they are located. A similar procedure was performed and validated by El-Fitiany and Youssef [6].

### 4.2. Circular Sections

To determine the temperature within the circular cross-section along the RC columns, the area is first divided into *M* concentric layers as shown in Figure 5b. The change in temperature (*T*) in each circular layer is derived by solving the heat balance equations at each time increment, assuming that the column is exposed to heat along its circumference, as described by Lie [16]. The influence of steel bars and moisture contents is considered in the analysis in a similar manner to the rectangular sections.

In this study, a method is proposed and validated to transform the circular layers into equivalent horizontal layers that can be utilized in the sectional analysis procedure. The procedure commences by dividing the semi-circular section into *M* horizontal layers (*I*), each corresponding to a unique circular layer (*J*), as indicated in Figure 5c. The upper and lower boundaries of any horizontal layer (*I*) are taken as the tangents to the two circular layers denoted by (*J* = *I*) and (*J* = *I* − 1), respectively. The intersection between the horizontal and circular layers produces elementary layers whose temperatures represent the temperature of the circular element they are located in. The area (*A*) of each elementary layer is derived in terms of the distance (*r*) from the center of the circular cross-section to each layer, as given in Equation (3).

$$A_{I,J} = \begin{cases} \dfrac{\pi\, r_1^2}{2} & , I = J = 1 \\[2ex] r_I^2 \times \dfrac{2\cos^{-1}\left(\frac{r_{I-1}}{r_I}\right) - \sin\left[2\cos^{-1}\left(\frac{r_{I-1}}{r_I}\right)\right]}{2} & , I = J \\[2ex] \dfrac{\pi\, r_J^2}{2} - r_J^2 \times \dfrac{2\cos^{-1}\left(\frac{r_I}{r_J}\right) - \sin\left[2\cos^{-1}\left(\frac{r_I}{r_J}\right)\right]}{2} - \sum_{1,1}^{I,J} A_{i,j} & , I \neq J \end{cases} \tag{3}$$

The temperature in each layer is calculated twice, similar to the procedure performed in rectangular sections. However, in the first case, the weighted average is calculated for each layer instead of calculating the normal average. This requires the determination of the area and temperature of each small element composing the horizontal layer. In the second case, the average temperature that would result in the same weighted average of residual compressive strength is determined. The temperature of each steel layer is taken as the maximum temperature reached at a distance equal to the provided concrete cover since all bars are uniformly distributed parallel to the circumference.

For both rectangular and circular columns, the temperature distribution within the section varies with the thermal properties of concrete and the cross-sectional dimensions. Figure 6 illustrates the change in temperature at different points along the mid-width of sections R3 and C3, whose characteristics are detailed in Table 1. The location of each point is defined as the distance from the face of the column in terms of section height (*h*) for rectangular sections and radius (*r*) for circular sections. Two main observations can be

drawn from these figures. Firstly, curves representing the points further away from the surface show a continuous increase in temperature after the end of heating. This causes the maximum temperature in the interior elements to be reached during the cooling phase, indicating that heat flow propagates not only to the atmosphere, but also to the inner, colder portions of the member. The second observation shows that cooling continues for a considerable amount of time before heat flow starts to take one direction only toward the atmosphere. A distinction between the rectangular and circular sections is detected in terms of response to temperature variation. In the aforementioned two sections, the concrete in column C3 located at a distance of up to (0.5 *r*) responds faster to the increase in temperature than that in rectangular sections located at the same distance. However, at a greater depth within the section, temperature variation becomes less pronounced in the circular section compared to its rectangular counterpart. This change in behavior is attributed to the more concrete area acting as a protecting cover for points closer to the core in section C3 compared to section R3.

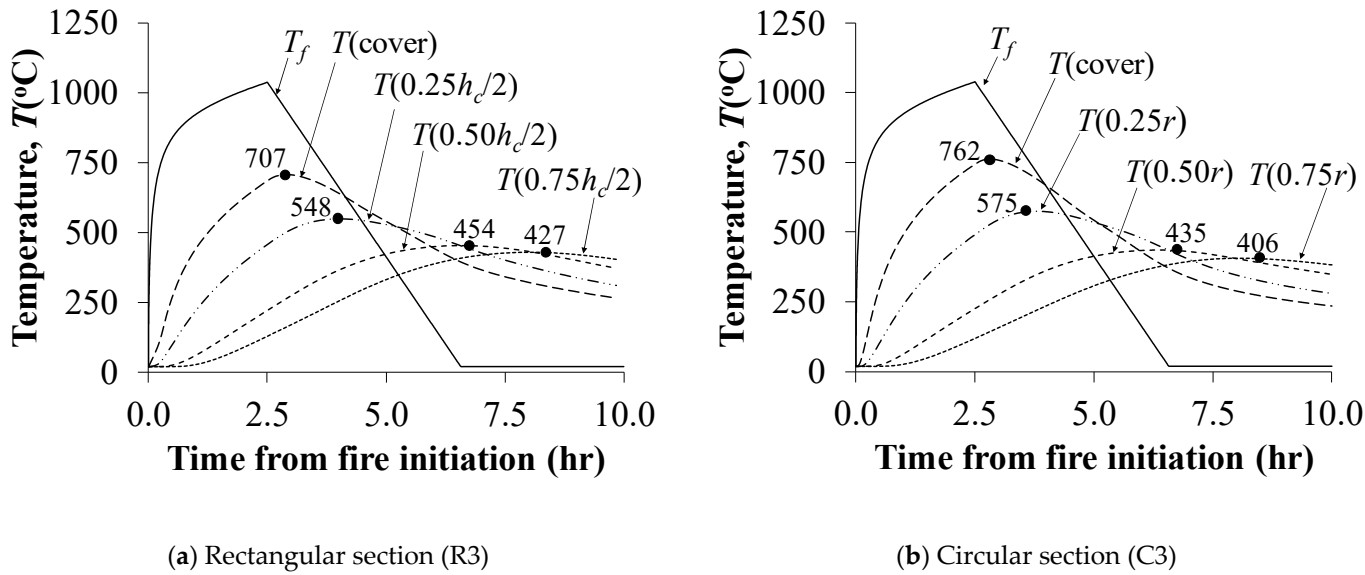

(**a**) Rectangular section (R3)　　　　　　　　　　　　　　　(**b**) Circular section (C3)

**Figure 6.** Temperature variation with time at different points along the cross-section.

Temperature distributions within sections R3 and C3 corresponding to maximum temperature reached as well as the end of both the heating and cooling phases are shown in Figures 7 and 8, respectively. As indicated in Figures 7a and 8a, heat flow is initiated from the section perimeter towards the inner core, resulting in the highest temperature rise near the exposed surfaces and the lowest values at the center point of section. During the gradual cooling phase, heat transfer takes place from the hot outer regions towards both the colder concrete zones and the surrounding air. This causes the temperature to keep increasing in the interior concrete elements for a certain period, beyond which heat transfer towards the atmosphere becomes predominant, as shown in Figures 7b and 8b, for the rectangular and circular sections, respectively. The maximum temperature distribution attained at each point within the section throughout the heating–cooling cycle is illustrated in Figures 7c and 8c for the same two sections, respectively. Maximum temperature distribution results in higher temperature values than those at the end of the heating phase. Hence, the residual mechanical properties and constitutive relationships of both concrete and steel are determined in the following sections based on the maximum temperature reached.

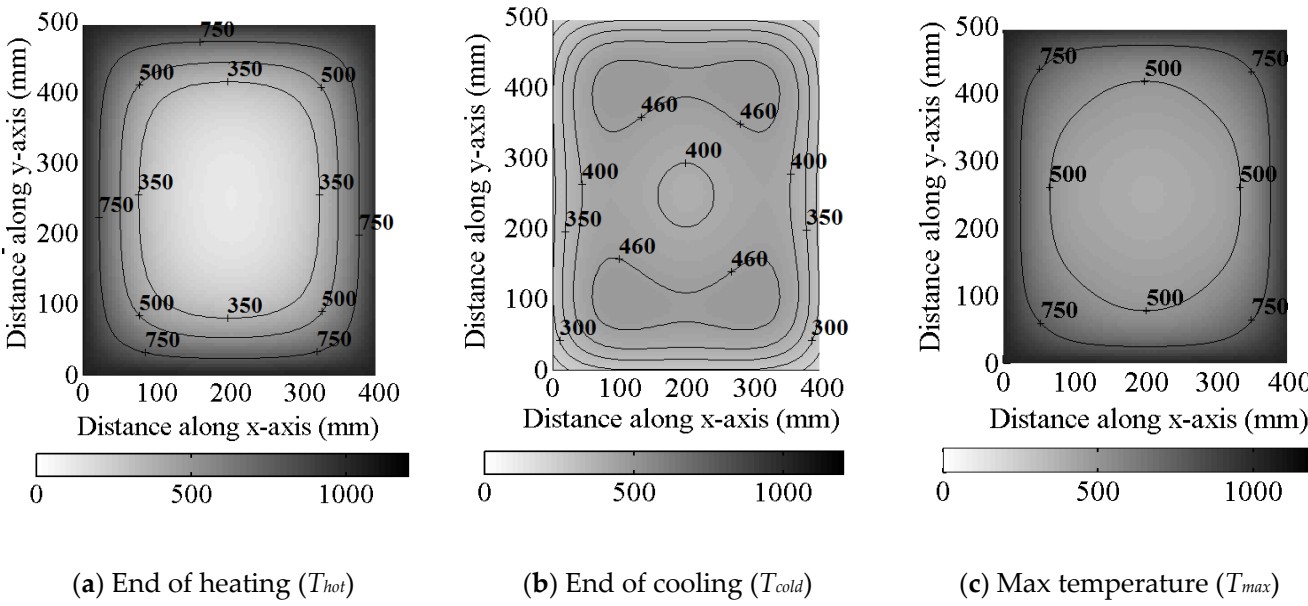

    (**a**) End of heating ($T_{hot}$)            (**b**) End of cooling ($T_{cold}$)           (**c**) Max temperature ($T_{max}$)

**Figure 7.** Temperature distribution within the rectangular cross-section of column (R3) at different time increments.

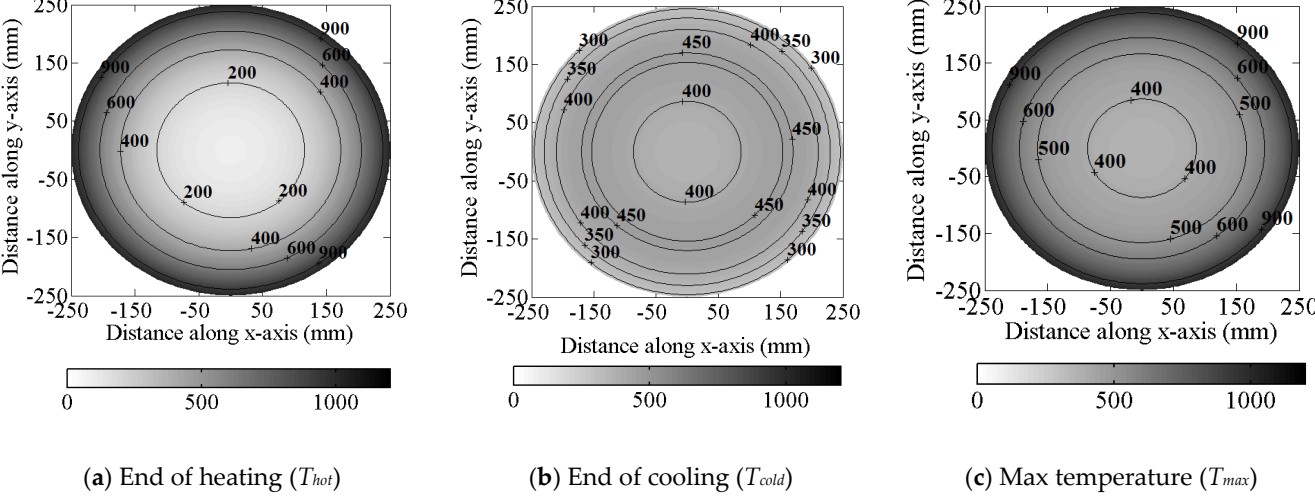

    (**a**) End of heating ($T_{hot}$)            (**b**) End of cooling ($T_{cold}$)           (**c**) Max temperature ($T_{max}$)

**Figure 8.** Temperature distribution within the circular cross-section of column (C3) at different time increments.

## 5. Material Models and Strain Components

The general form of the Tsai [19] model is adopted in this study to represent the compressive stress–strain relationship of concrete at all stages. During fire, the reduced compressive strength due to fire ($f'_{cT}$) proposed by Hertz [15] is used, whereas concrete strain at peak stress at elevated temperatures ($\varepsilon_{oT}$) is determined by the Terro [20] formula. The post-fire mechanical properties are calculated based on the expressions by Chang et al. [21].

Regarding steel, the constitutive model used by Karthik and Mander [22] is adopted for both ambient and post-fire conditions as it conveniently combines the initial elastic response, yield plateau and strain hardening stages. At elevated temperatures, the Lie [23] model is used as it implicitly includes the reduction in yield strength due to fire.

Total strain in concrete ($\varepsilon_t$) is calculated as the summation of stress-related strain ($\varepsilon_\sigma$), free thermal strain ($\varepsilon_{th}$), creep strain ($\varepsilon_{cr}$), and transient strain ($\varepsilon_{tr}$). The tendency of the structural members to deform due to external applied loads is described in terms of the

stress-related strain component. Free thermal strain of both concrete and steel bars is determined from Eurocode [4]-proposed expressions. The residual free thermal strain ($\varepsilon_{thR}$) represents the irreversible part of the free expansion that occurred during fire. After a complete heating–cooling cycle, thermal strain is restored with a rate of $8 \times 10^{-6}/°C$ from the maximum temperature reached [1], while $\varepsilon_{thR}$ for steel is set to zero. If the member is initially loaded or restrained, then transient strain is generated in concrete and maintains its maximum values after cooling [1]. The empirical model proposed by Terro [20] is adopted to calculate the transient creep strain as referred to by load-induced thermal strain ($\varepsilon_{LITS}$). Regarding steel bars, the residual thermal strain is brought back to zero at the end of the cooling phase. Both transient and creep strain are not applicable for steel during and after fire. Detailed descriptions of the material models and strain components during fire exposure are provided by Youssef and Moftah [5].

## 6. Strength Analysis

An iterative sectional analysis procedure is carried out to determine the residual *P-ε* behavior of the fire-damaged RC columns. The residual properties are determined in view of the temperature distribution obtained from thermal analysis. At every loading step, the axial strain is increased incrementally until reaching the total applied axial load. The kinematic and compatibility conditions are considered in view of the corresponding residual mechanical properties and stress–strain relationships of both concrete and steel. The strength analysis is performed by dividing the cross-section into multiple horizontal layers, as shown in Figures 4c and 5d, for the rectangular and circular cross-sections, respectively. To maintain the high accuracy while reducing the computation time, a sensitivity analysis was performed, and the maximum layer height was chosen as not to exceed 3 mm. The centroid of each concrete and steel layer is determined considering the appropriate geometrical expressions for both circular and rectangular sections. For concrete, temperature is obtained from the average distribution that would result in average compressive strength in each layer, whereas the maximum temperature reached is used directly for steel layers corresponding to the exact location of steel bars. The failure criterion of the RC element is defined by the crushing of concrete once the strain in any of the sectional layers reaches the residual ultimate strain ($\varepsilon_{cuR}$) proposed and validated by Alhadid and Youssef [24]. The restraining effect due to elevated temperature is considered in the analysis through calculating the axial restraint at each time increment depending on the assumed supporting condition. The axial force generated due to restraint is added to the initial applied load to determine the total axial load during fire exposure.

## 7. Equivalent Residual Strain

Residual stresses are induced in fire-damaged members for two main reasons:

(1) Thermal strain in concrete is partially reversible, while transient strain is completely irreversible [1]. At equilibrium, unloaded fire-damaged concrete tends to remain either expanded or contracted depending on the temperature–load history. On the other hand, thermal strain in steel is fully reversible and the embedded steel bars tend to restore their initial length after fire, provided that they did not reach the yield point at the elevated temperature. The variation in behavior between concrete and the embedded steel bars generates internal stresses.

(2) Both thermal and transient strain distributions along section height are nonlinear as they follow the nonlinear temperature profile. Therefore, internal stresses are developed in order to maintain the plane section assumption.

Figure 9 illustrates the development of the strain components along section (A-A) of Figure 4c for rectangular sections. The same analysis procedure is considered for circular sections while accounting for the modified location of the steel layers. The difference between the residual thermal strain ($\varepsilon_{thR}$) and the residual transient strain ($\varepsilon_{trR}$) is the total residual strain ($\varepsilon_R$), which can be either positive or negative depending on the temperature–load history and the magnitude of the developed transient strain. Due to the plane section

assumption, the deformed section is represented by a uniform equivalent strain ($\varepsilon_{eq}$) along the cross-section. Residual stress-induced strain ($\varepsilon_{\sigma i}$) distribution is determined as the difference between an equivalent strain ($\varepsilon_{eq}$) and the total residual strain ($\varepsilon_R$). An iteration process is performed to evaluate the uniformly distributed equivalent strain ($\varepsilon_{eq}$) that satisfies the equilibrium condition of $\varepsilon_{\sigma i}$ distribution. The value of $\varepsilon_{eq}$ is determined such that the total axial force in concrete and steel resulting from $\varepsilon_{\sigma i}$ distribution is equal to zero.

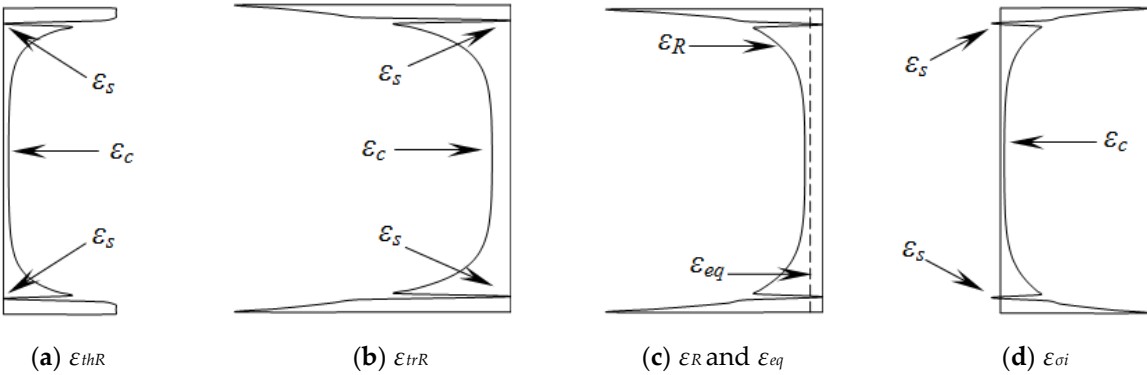

**(a)** $\varepsilon_{thR}$      **(b)** $\varepsilon_{trR}$      **(c)** $\varepsilon_R$ and $\varepsilon_{eq}$      **(d)** $\varepsilon_{\sigma i}$

**Figure 9.** Development of various strain components along the discretized cross-section.

Once equilibrium is achieved, $\varepsilon_{\sigma i}$ is applied as initial strains in the concrete and steel layers, whereas $\varepsilon_{eq}$ results in shifting the *P-$\varepsilon$* curve, as illustrated in Figure 10 for both rectangular and circular sections. The residual and equivalent strain distribution along column R3 and C3 cross-sections are shown in Figure 11. If the column is not initially loaded during fire exposure ($\lambda = 0$), then the residual equivalent strain ($\varepsilon_{eq1}$) is always negative, causing the *P-$\varepsilon$* curve to shift to the expansion side. However, by imposing an initial load to the column during the heating phase, a transient strain component develops and counteracts the influence of the thermal strain. If the applied load is large enough, the column experiences residual contraction instead of expansion after the cooling, as indicated by the positive equivalent strain ($\varepsilon_{eq2}$). The change in stiffness is attributed to the elimination of the residual stress-induced strains. Restraining the column affects the magnitude of the generated transient strain, especially if the column is not subjected to initial load. When the column is restrained, part of the equivalent strain ($\varepsilon_{eq}$) induces stresses within the section depending on the considered degree of restraint while maintaining the equilibrium condition. By restraining the column, additional compressive forces are developed in the column as a result of preventing the column's tendency to expand.

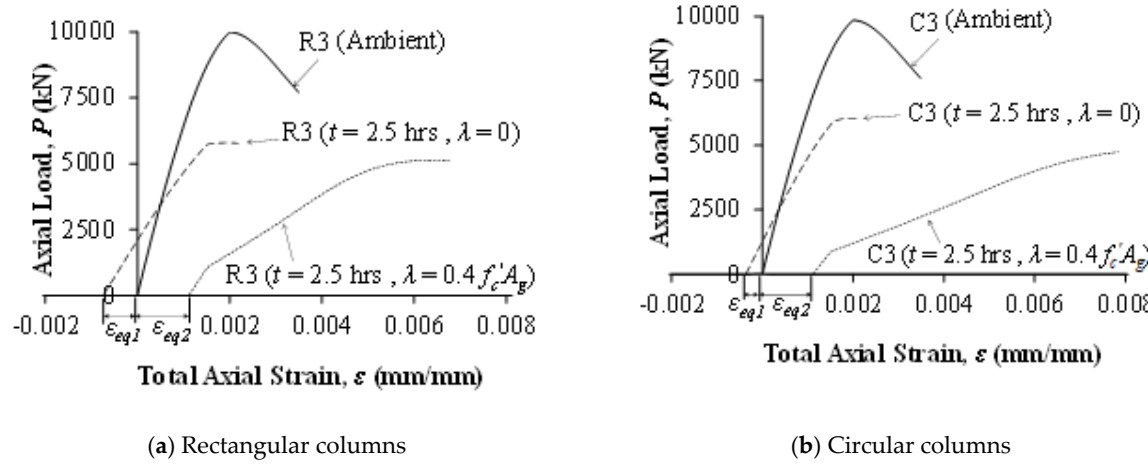

**(a)** Rectangular columns      **(b)** Circular columns

**Figure 10.** Influence of initial load level on the residual (*P-$\varepsilon$*) relationship.

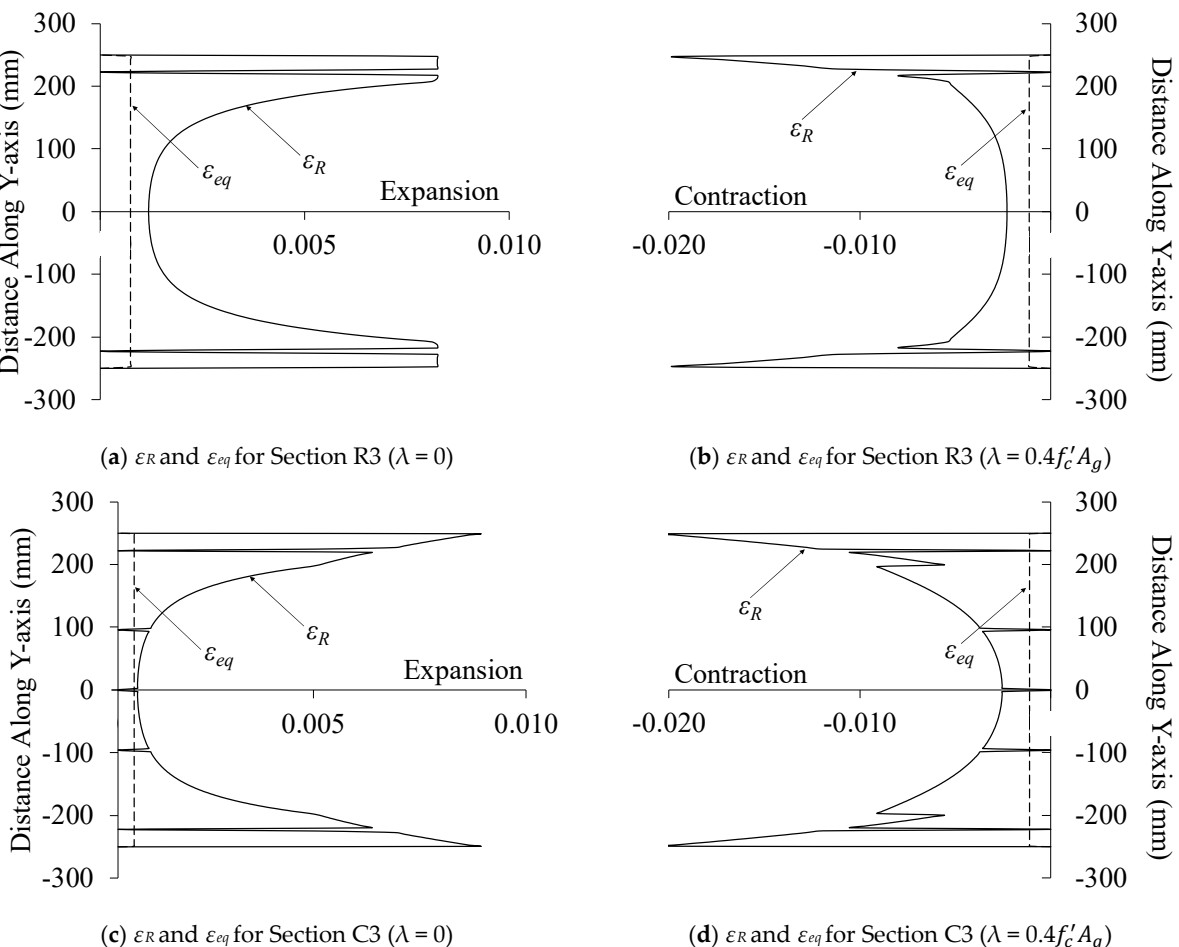

(**a**) $\varepsilon_R$ and $\varepsilon_{eq}$ for Section R3 ($\lambda = 0$)

(**b**) $\varepsilon_R$ and $\varepsilon_{eq}$ for Section R3 ($\lambda = 0.4f_c'A_g$)

(**c**) $\varepsilon_R$ and $\varepsilon_{eq}$ for Section C3 ($\lambda = 0$)

(**d**) $\varepsilon_R$ and $\varepsilon_{eq}$ for Section C3 ($\lambda = 0.4f_c'A_g$)

**Figure 11.** Residual and equivalent strain distribution along column R3 and C3 cross-sections.

## 8. Validation of the Proposed Analytical Model

The capability of the present model to predict the post-fire structural performance of axially loaded RC members is validated in view of the experimental results by Chen et al. [25], Jau and Huang [26], Yaqub and Bailey [27] and Elsanadedy et al. [28]. The validation is limited to structural members made of normal-strength concrete where spalling does not occur.

Chen et al. [25] carried out a full-scale experiment to investigate the performance of RC columns after exposure to different fire conditions. The results obtained from the proposed analytical model are compared with the measured data of columns FC06 and FC05. These columns are exposed to an ISO 834 (2014) standard fire curve from four sides for 2 hrs and 4 hrs, respectively. The tested columns have cross-sectional dimensions of 300 mm × 450 mm, concrete cover of 40 mm and an overall length of 3.0 m. The concrete compressive strength at ambient conditions is 29.5 MPa. The longitudinal reinforcement consists of 4 Φ 19 mm and 4 Φ 16 mm steel bars with yield strengths of 476 MPa and 479 MPa, respectively. Both columns were subjected to an initial axial load of 797 kN prior to heat exposure. The specimens were axially loaded during the whole heating and cooling cycle. After 30 days from the fire test, the columns were subjected to a constant initial concentric load of 797 kN and an additional eccentric load offset a distance of 650 mm from the cross-sectional centre along the y-axis (weak axis) producing the bending moment about the x-axis, while another eccentric load is applied at 600 mm from the centroidal axis. Figure 12a shows the analytical and experimental load–deflection curves at the column mid-span due to the eccentric load about the y-axis. A very good agreement between both curves can be shown with a percent difference of 3.8% and 4.6% in the ultimate capacity of columns FC06 and FC05, respectively, and a percent difference of 6.3% and 5.4% in the 40%

secant stiffness for the same two columns, respectively. This variation can be attributed to the sensitivity of the adopted thermal expansion model to the experimental conditions and concrete mix.

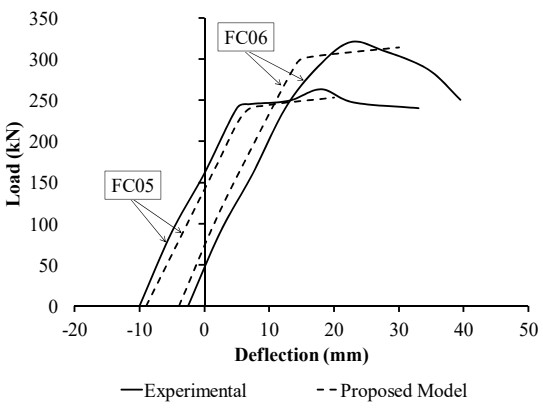

(**a**) Using experiment by Chen et al. (2009)

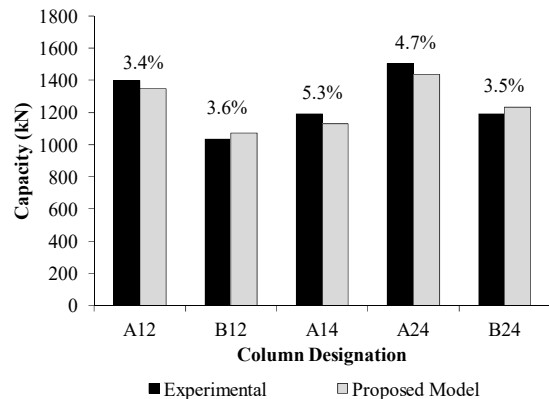

(**b**) Using experiment by Jau and Huang (2008)

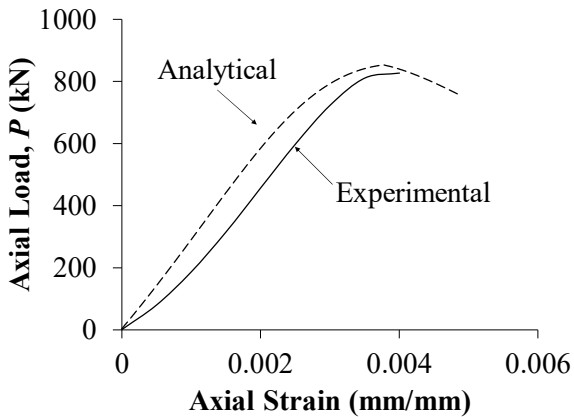

(**c**) Using experiment by Yaqub and Bailey (2011)

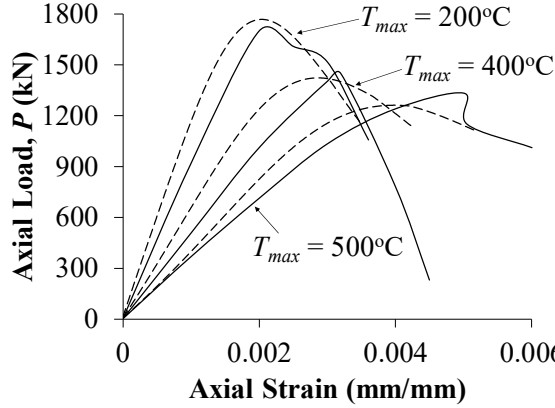

(**d**) Using experiment by Elsanadedy et al., (2016)

**Figure 12.** Validation of the proposed analytical model.

In another experimental study, Jau and Huang [26] investigated the residual behavior of initially loaded restrained RC columns subjected to heat from two adjacent sides. The cross-sectional dimensions of all columns are 300 mm × 450 mm, with an overall length of 2.7 m. The concrete cover varies between 50 mm or 70 mm, whereas the steel reinforcement ratio varies between 1.8% and 3.0%. Normal-strength concrete with a compressive strength of 33.7 MPa and steel bars with a yield strength of 475.8 MPa are used. The test setup allows the heat to flow through two adjacent surfaces only while the other two surfaces are insulated and not subjected to fire. The restrained columns are subjected to a 10% axial preloading of their ambient compressive strength during the 2 or 4 hr fire tests. After the columns naturally cooled down, the load is applied until failure occurs. Figure 12b shows both the experimental and predicted residual capacity of columns A12, B12, A14, A24 and B24 whose detailed geometrical and mechanical properties are provided by Jau and Huang [26]. The proposed model is found to predict the capacity of the tested columns with high accuracy, as indicated by the maximum percent error of 5.3% depicted for column A14 shown in Figure 12b. Overall, the agreement between the experimental and analytical results is very good. It is worth mentioning that exposing the specimens to fire from two adjacent sides only resulted in a lateral displacement in the range of 7 to 25 mm, depending on fire duration. Due to the influence of load–temperature interaction, the residual strength

and stiffness varies within the cross-section. The lateral displacement and the associated curvature should be considered when analyzing the fire-exposed columns, especially those with high slenderness ratios.

The influence of elevated temperature on the residual axial capacity, axial stiffness and stress–strain behavior of circular columns strengthened with fiber-reinforced polymers (FRP) was experimentally investigated by Yaqub and Bailey [27]. The unwrapped control specimen (i.e., Specimen No. 2) is considered for comparison. The examined column has a diameter of 200 mm and an overall length of 1000 mm. The concrete cover to the centroid of the steel bars was taken as 30 mm. The reinforcement consisted of 6 Φ 10 mm steel bars, resulting in a reinforcement ratio of 1.5%. Normal-weight concrete with a compressive strength of 42.4 MPa and steel bars with a yield strength of 570 MPa were used. The column was exposed to a predefined heating–cooling cycle 9 months after casting until the entire cross-section reached a uniform temperature of 500 °C. After that, the column was subjected to a displacement-controlled uniaxial compression load until failure. The member temperature at the time of the compression test was 22 °C. Figure 12c presents both the experimental and analytical axial load–deformation curves for the specimen, which was exposed to a uniform temperature of 500 °C before cooling. The proposed model is found to provide very good prediction of the experimental results, as indicated by the 4.2% percent error. The strength of the heat-exposed columns was reduced by 41.8% after the heating–cooling cycle, as implied by the strength reduction from an average of 1418 kN for the intact columns to 826 kN of the heated column. Based on Figure 4.4.2.2.1c(b) of ACI 216-14 [9], the estimated residual strength of the specimen at 500 °C is determined as about 51.0% of the initial compressive strength. This represents an error of about 12.4% with respect to the actual residual strength of 58.2% determined in the lab. The incremental stiffness at service load is almost identical between the two curves. Additionally, the load–deformation behavior obtained from the proposed model is shown to be consistent with that obtained experimentally in terms of stiffness, peak strain and failure strain.

Elsanadedy et al. [28] examined the effect of high temperature on the residual capacity and deformation behavior of R.C. columns strengthened with FRP wraps. The control specimens, which were unwrapped, were tested at a room temperature of 26 °C and are considered for comparison in this paper. All the examined columns have a diameter of 242 mm and an overall length of 900 mm. The concrete cover to the centroid of the steel bars was taken as 41 mm. The reinforcement consisted of 4 Φ 10 steel bars, resulting in a reinforcement ratio of 0.68%. Normal-weight concrete with a compressive strength of 42 MPa and steel bars with a yield strength of 593 MPa were used. The columns were exposed to elevated temperature along their circumference under unstressed conditions with a heating rate ranging between 5 °C and 15 °C per minute. Specimens C-200, C-400 and C-500 are considered for comparison, where the letter "C" indicates un-strengthened specimens, and the number indicates the maximum temperature reached in the oven. The specimens were subjected to the specified maximum temperature for 3 h before shutting down the oven. The columns were then naturally cooled inside the oven to room temperature. After that, the columns were taken out of the oven and subjected to a displacement-controlled uniaxial compression load until failure. Figure 12d presents both the experimental and analytical axial load–deformation curves for the examined specimens. The residual strength of specimens C-200, C-400 and C-500 were found to be 1745 kN, 1490 kN and 1350 kN, respectively. These values represent a residual strength of 90.1%, 77.1% and 69.9% of the initial strength of the intact specimen, respectively. The capability of the proposed model to capture the residual capacity of the heat-exposed columns is very good, as indicated by the 4.7%, 3.7% and 6.5% percent errors for specimens C-200, C-400 and C-500, respectively. Considering Figure 4.4.2.2.1c(b) of ACI 216-14 [9], the residual strength of specimens C-200, C-300 and C-400 are 84.0%, 75.0% and 63.0% of the initial compressive strength, respectively. This represents a percent error of 7.0%, 2.8% and 9.8% relative to the tested specimens, respectively. Additionally, the load–deformation behavior obtained from the proposed model is shown to be consistent with that obtained experimentally in terms of stiffness,

peak strain and failure strain. The error between the model and experimental results can be attributed to the variation in heating rate and the presence of residual surface cracks and initial misalignment, which are not accounted for in the model.

## 9. Parametric Study

The main parameters include the concrete compressive strength, $f_c'$ (25 MPa and 35 MPa); steel yield strength, $f_y$ (300 MPa and 400 MPa); fire duration, $t$ (0.5 hr, 1.5 hrs and 2.5 hrs); initial load level, $\lambda$ (0.0, 0.2 $f_c'$, 0.4 $f_c'$); axial restraint stiffness ratio, $R_D$ (0.0, 0.5 and 1.0); and steel reinforcement ratio, $\rho$ (0.02 and 0.04). The cross-sectional dimensions of the rectangular sections are defined in terms of member height, $h$ (400 mm and 800 mm) and width, $b$ (300 mm and 600 mm), whereas for circular sections, the geometrical properties are determined in terms of their diameter, $D$ (350 mm and 650 mm). A 30 mm clear concrete cover was considered for all specimens. The members are exposed to fire along their perimeters according to ASTM E119 [17] standard fire curve, followed by a cooling phase according to ISO 834 [18] recommendations. The influence of the considered factors on the post-fire behavior of both rectangular and circular RC axially loaded members is investigated in view of a parametric study. Based on these parameters, the analytical investigation consists of a total of 1728 different cases.

The effect of the aforementioned parameters on both the residual axial capacity and the residual 40% secant axial stiffness is illustrated in view of the members presented in Table 1. The variation in the residual capacity and stiffness in terms of the different parameters at different initial load levels is presented Figures 13 and 14 for both rectangular and circular sections, respectively.

### 9.1. Effect of Fire Duration

Fire duration has been found to have the most significant influence on reducing the post-fire capacity and stiffness of both rectangular and circular RC columns. The influence of increasing the fire duration on the residual flexural behavior is examined in view of the rectangular sections (R1, R2 and R3) and the circular sections (C1, C2 and C3), as shown in Figures 13a and 14a, respectively. Prolonged exposure to fire results in material strength degradation and softening, which adversely affect the stiffness and capacity of the fire-damaged section. The permanent strength and stiffness reductions in the circular columns are found to be slightly higher than those with rectangular sections. This can be attributed to the higher maximum temperature reached within the circular sections subjected to fire for the same fire duration, as was previously described in Figure 6. The additional deterioration in both concrete and steel residual mechanical properties caused by the longer duration of the heating–cooling cycle provides more time for heat to transfer to the inner elementary layers raising their temperatures.

### 9.2. Effect of Section Size

Increasing the cross-sectional dimensions of both rectangular and circular columns results in higher residual flexural strength and stiffness after fire, as indicated in Figures 13b and 14b. This larger residual capacity is caused by the lower temperature increase within the larger member as it requires more heat energy to increase its temperature. This is attributed to the additional concrete cover provided by the larger sections causing the hindrance of heat transfer from the column perimeter towards its core. Hence, internal concrete fibers experience lower temperatures and consequently higher residual compressive strength and stiffness than the inner elements of columns with smaller dimensions. For the same fire duration, concrete within the inner parts of the wider member experience a lower increase in temperature and consequently more recovery after a fire. The influence of strength recovery in steel bars is neglected since concrete cover is the same in all specimens causing the maximum temperature reached in all steel bars to be the same.

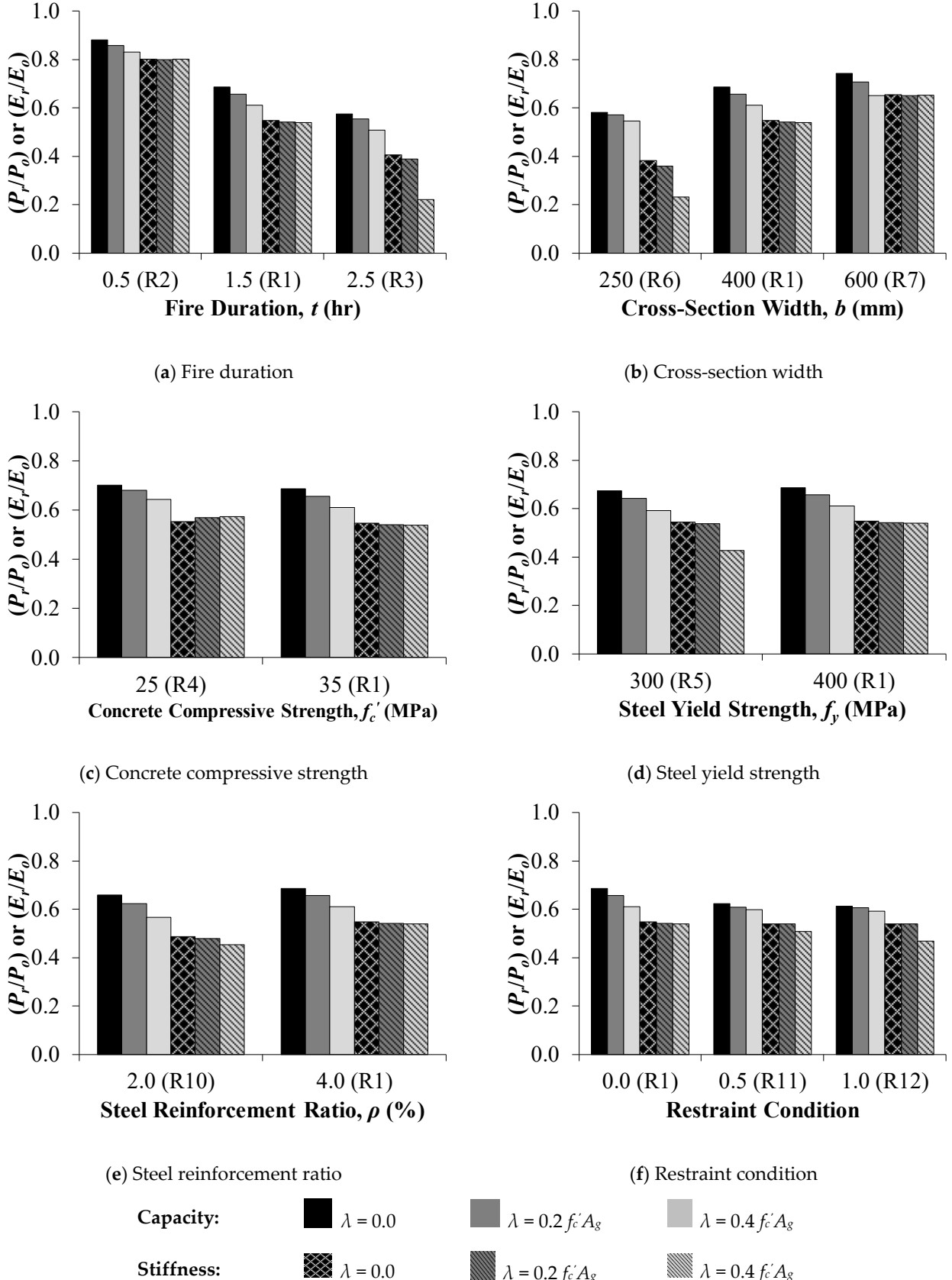

**Figure 13.** Influence of varying the examined parameters on the axial capacity and stiffness of rectangular columns.

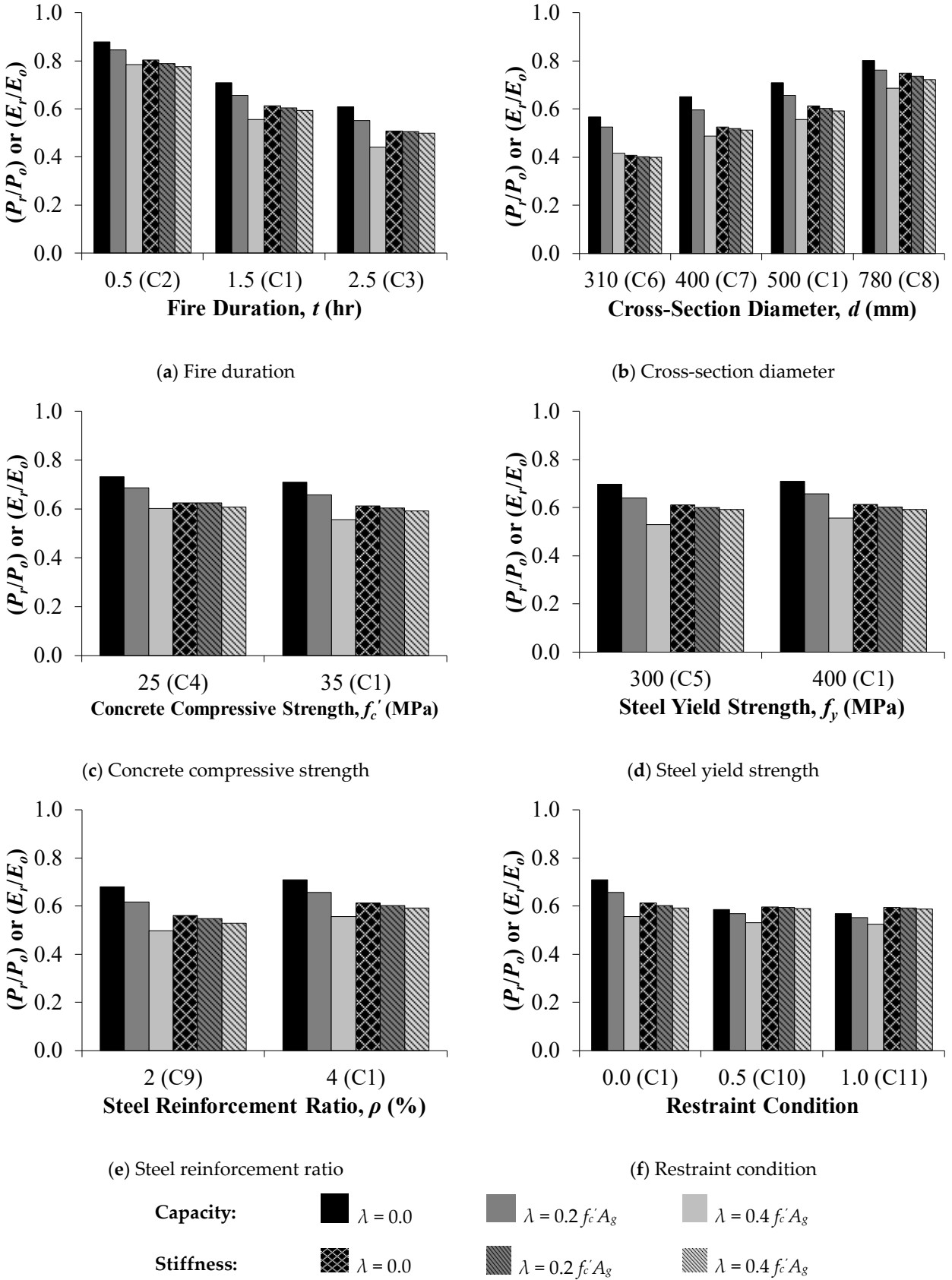

**Figure 14.** Influence of varying the examined parameters on the axial capacity and stiffness of circular columns.

*9.3. Effect of Mechanical Properties*

Increasing the concrete compressive strength is found to have an insignificant inverse relationship on the reduction ratio of both capacity and stiffness for all load levels in the examined range, as shown in Figures 13c and 14c, for rectangular and circular columns, respectively. The decreasing rate can be justified by the greater reduction in compressive strength of the stronger concrete after fire. Hence, the reduction in concrete contribution within the compression zone becomes more pronounced and results in the observed larger decrease relative to the original capacity. The use of normal-strength concrete infers that no spalling is encountered, which could otherwise significantly affect the residual capacity. The same observation can be drawn by varying the grade of the embedded steel bars from 300 MPa to 400 MPa, as shown Figures 13d and 14d, for rectangular and circular columns, respectively. This is attributed to the fact the steel bars restore a significant portion of their capacity and stiffness after fire as discussed previously.

*9.4. Effect of Steel Reinforcement Ratio*

Steel bars are located near the exposed surfaces of the columns and are subjected to relatively high temperatures. However, this has negligible impact on the overall axial capacity and stiffness reduction due to the significant recovery of mild steel bars after fire exposure [29–31]. Figures 13e and 14e show that increasing the reinforcement ratio results in an insignificant increase in both residual capacity and stiffness in the rectangular and circular columns, respectively. This is attributed to the higher impact of the larger steel area in replacing the fire-damaged concrete since the recovery of steel bars is very significant as opposed to concrete.

*9.5. Effect of Restraint Conditions*

The influence of restraining the member against thermal expansion during heating has been found to slightly decrease its post-fire stiffness and capacity, as shown in Figures 13f and 14f, for both rectangular and circular columns, respectively. The reduction in residual properties is more pronounced when comparing the fully unrestrained sections with the restrained ones. However, the reduction seems to be almost identical for columns that are fully restrained or 50% restrained. This is explained by the impact of transient strain in changing the deformation behavior of axially loaded members during fire exposure through alleviating the thermal expansion. As the stiffness of the supports provided by the adjacent frame members increases, more restraining forces are generated to counteract the tendency of the column to expand. This additional force results in transient creep strain, which reduces the thermal strain and consequently decreases the amount of restraining force required to overcome the expansion. These two processes occur simultaneously and have a negative influence on each other, causing them to reduce the impact of restrains.

During fire exposure, the column's tendency to undergo thermal expansion increases with time, causing the support to counteract this potential movement, depending on the column's stiffness. Initially, the member's stiffness remains close to that at ambient conditions as the temperature increase within the member is relatively low. Thus, an increase in restraining force results in significant hindrance of the column's deformation as the thermal strain component increases. However, after a certain period of time, the temperature within the member becomes relatively high, causing the stiffness degradation to become more pronounced. Thus, the forces required to resist the larger thermal expansion of the member drops. The axial force required to restrain the member keeps decreasing as a result of the continuous reduction in stiffness caused by elevated temperatures. Therefore, the change in the restraining load is characterized by a mild increase followed by a gradual decrease with time.

## 10. Proposed Simplified Expressions to Obtain Residual Axial Capacity and Stiffness

Prolonged exposure of RC columns to elevated temperatures according to a standard fire has a substantial influence on their axial capacity and deformation behavior.

The residual structural performance of such columns relies on the geometrical characteristics, mechanical properties, initial load, restraint conditions and fire duration that should be appropriately accounted for in the analysis. Accurate determination of temperature distribution and residual strain components developed within RC columns is tedious and requires detailed thermal and structural analyses that may not be convenient for design engineers. The proposed analytical model comprehensively addresses the influence of the aforementioned factors on determining the post-fire response of both rectangular and circular RC columns. Hence, based on the extensive parametric study conducted on the 1728 different cases, regression analysis is carried out to develop expressions for obtaining both the residual axial capacity and secant axial stiffness of fire-damaged rectangular and circular RC columns. These proposed expressions take into consideration the loading history, restraint conditions, fire duration, material strength and cross-sectional dimensions of the exposed members. The validity and accuracy of the proposed equations depend on the range of parameters considered in the parametric study. The proposed expressions provide a suitable approach for predicting the behavior of RC columns after exposure to an extreme standard fire scenario. This would be a valuable tool for both researchers and engineers to predict the post-fire performance of RC columns during the design phase.

### 10.1. Rectangular Sections

Linear multiple regression analysis is performed to propose an expression for both the residual capacity and axial stiffness ratios ($\omega$), as given in Equation (4).

$$\omega = A_1 + A_2\lambda + A_3f_c' + A_4f_y + A_5\rho + A_6\frac{\rho f_y}{f_c'} + A_7b + A_8h \tag{4}$$

where $\lambda$ is the initial load level relative to ambient capacity, $f_c'$ is the concrete compressive strength (MPa), $f_y$ is the steel yield strength (MPa), $\rho$ is the steel reinforcement ratio, $b$ is section width (m), and $h$ is section height (m). The coefficients ($A_{i = 1, 2, 3, 4, 5, 6, 7, 8}$) are given in Table 2 in terms of the axial restraint ratio ($R_D$) and fire duration at the end of the heating phase ($t$) in hours. For values other than the listed $t$ and $R_D$, linear interpolation of the upper and lower calculated $\omega$ should be performed. In Table 2, $P_o$ and $P_r$ are the axial capacities at ambient and post-fire conditions, respectively; $EA_i$ and $(EA_i)_r$ are the initial axial stiffness at ambient and post-fire conditions, respectively; $EA_{0.4}$ and $(EA_{0.4})_r$ are the 40% axial stiffness at ambient and post-fire conditions, respectively; and $EA_{0.8}$ and $(EA_{0.8})_r$ are the 80% axial stiffness at ambient and post-fire conditions, respectively.

It is worth mentioning that although the rectangular column is exposed to fire from all sides, the coefficients of the section height ($h$) and section width ($b$) are different in Equation (4). This variation is attributed to the assumed reinforcement configuration where the steel bars lie in two opposite layers that are parallel to the section width as indicated in Figure 4a.

The applicability of the proposed expressions is assessed by comparing the values obtained using the proposed equations and the results obtained from the analytical analysis. A comparison between the values predicted from Equation (4) and the results determined through performing detailed analytical analysis for all examined cases revealed a very good agreement, as shown in Figures 15a and 16a, for both residual capacity and axial stiffness, respectively. The equality line denotes the location on the graph where the predictions from the proposed equations match those obtained from the proposed analytical model. As shown in the figure, the data points are uniformly distributed in the vicinity of the equality line.

The residual compressive strength of column R6 is determined from ACI 216.1-14 and compared to that obtained from Equation (4) considering an exposure of 1 h to ASTM E119 [17] standard fire. Figure 4.4.2.3a and 4.4.2.2.1c(b) of ACI 216.1-14 provide the temperature distribution and the residual compressive strength of concrete at various heating and loading conditions, respectively. The unstressed residual compressive strength obtained from the ACI procedure for column R6 is found to be just under 71.0% of its initial strength.

Considering the same parameters, the percentage of the residual strength obtained from the proposed Equation (4) is found to be 77.7%. In general, there is a good agreement between the two values considering the complex behavior of fire-exposed RC members. The variation between the two values is attributed mainly to the difference between the exposure and boundary conditions assumed in both methods. The residual strength is calculated from Equation (4) considering the reinforcement ratio, steel yield strength and section height, which is not specifically provided in the ACI procedure.

**Table 2.** Coefficient of Equation (4) for rectangular sections.

| $\omega$ | $A_i$ | $R_D = 0$ (Unrestrained) | | | $R_D = 0.5$ (Partially Restrained) | | | $R_D = 1$ (Fully Restrained) | | |
|---|---|---|---|---|---|---|---|---|---|---|
| | | $t = 0.5$ hr | $t = 1.5$ hrs | $t = 2.5$ hrs | $t = 0.5$ hr | $t = 1.5$ hrs | $t = 2.5$ hrs | $t = 0.5$ hr | $t = 1.5$ hrs | $t = 2.5$ hrs |
| $\frac{P_r}{P_o}$ | $A_1$ | $6.90 \times 10^{-1}$ | $3.60 \times 10^{-1}$ | $2.06 \times 10^{-1}$ | $6.16 \times 10^{-1}$ | $3.28 \times 10^{-1}$ | $2.17 \times 10^{-1}$ | $5.96 \times 10^{-1}$ | $3.14 \times 10^{-1}$ | $2.18 \times 10^{-1}$ |
| | $A_2$ | $-1.22 \times 10^{-1}$ | $-1.70 \times 10^{-1}$ | $-1.48 \times 10^{-1}$ | $-7.29 \times 10^{-2}$ | $-7.23 \times 10^{-2}$ | $-8.24 \times 10^{-2}$ | $-6.74 \times 10^{-2}$ | $-6.43 \times 10^{-2}$ | $-7.03 \times 10^{-2}$ |
| | $A_3$ | $-7.09 \times 10^{-4}$ | $-1.28 \times 10^{-3}$ | $-1.58 \times 10^{-3}$ | $-1.08 \times 10^{-3}$ | $-1.97 \times 10^{-3}$ | $-2.16 \times 10^{-3}$ | $-1.16 \times 10^{-3}$ | $-1.94 \times 10^{-3}$ | $-2.26 \times 10^{-3}$ |
| | $A_4$ | $5.01 \times 10^{-5}$ | $6.72 \times 10^{-5}$ | $7.83 \times 10^{-5}$ | $6.94 \times 10^{-5}$ | $8.57 \times 10^{-5}$ | $1.02 \times 10^{-4}$ | $7.58 \times 10^{-5}$ | $9.78 \times 10^{-5}$ | $1.04 \times 10^{-4}$ |
| | $A_5$ | 1.31 | 1.90 | 2.34 | 1.82 | 2.46 | 3.05 | 2.09 | 2.65 | 3.17 |
| | $A_6$ | $9.03 \times 10^{-2}$ | $1.16 \times 10^{-1}$ | $1.43 \times 10^{-1}$ | $1.14 \times 10^{-1}$ | $1.43 \times 10^{-1}$ | $1.41 \times 10^{-1}$ | $1.14 \times 10^{-1}$ | $1.42 \times 10^{-1}$ | $1.47 \times 10^{-1}$ |
| | $A_7$ | $1.66 \times 10^{-1}$ | $3.45 \times 10^{-1}$ | $4.04 \times 10^{-1}$ | $1.54 \times 10^{-1}$ | $2.51 \times 10^{-1}$ | $3.00 \times 10^{-1}$ | $1.49 \times 10^{-1}$ | $2.37 \times 10^{-1}$ | $2.78 \times 10^{-1}$ |
| | $A_8$ | $1.30 \times 10^{-1}$ | $2.23 \times 10^{-1}$ | $2.42 \times 10^{-1}$ | $1.78 \times 10^{-1}$ | $2.56 \times 10^{-1}$ | $2.43 \times 10^{-1}$ | $1.92 \times 10^{-1}$ | $2.61 \times 10^{-1}$ | $2.43 \times 10^{-1}$ |
| $\frac{(EA_i)_r}{EA_i}$ | $A_1$ | $5.36 \times 10^{-1}$ | $-1.38 \times 10^{-1}$ | $-3.18 \times 10^{-1}$ | $5.34 \times 10^{-1}$ | $-1.64 \times 10^{-1}$ | $-2.57 \times 10^{-1}$ | $5.32 \times 10^{-1}$ | $-2.41 \times 10^{-1}$ | $-2.75 \times 10^{-1}$ |
| | $A_2$ | $-7.45 \times 10^{-3}$ | $-5.66 \times 10^{-2}$ | $-7.28 \times 10^{-2}$ | $-3.75 \times 10^{-3}$ | $-6.59 \times 10^{-2}$ | $-1.26 \times 10^{-1}$ | $-3.55 \times 10^{-3}$ | $-8.61 \times 10^{-2}$ | $-1.35 \times 10^{-1}$ |
| | $A_3$ | $-3.90 \times 10^{-4}$ | $-2.01 \times 10^{-3}$ | $-3.58 \times 10^{-3}$ | $-4.00 \times 10^{-4}$ | $-3.97 \times 10^{-3}$ | $-2.77 \times 10^{-3}$ | $-3.56 \times 10^{-4}$ | $-2.69 \times 10^{-3}$ | $-4.56 \times 10^{-3}$ |
| | $A_4$ | $-2.41 \times 10^{-5}$ | $1.79 \times 10^{-4}$ | $3.70 \times 10^{-4}$ | $-2.83 \times 10^{-5}$ | $3.14 \times 10^{-4}$ | $2.02 \times 10^{-4}$ | $-2.98 \times 10^{-5}$ | $2.99 \times 10^{-4}$ | $3.83 \times 10^{-4}$ |
| | $A_5$ | 2.52 | 9.00 | $1.49 \times 10^{+1}$ | 2.55 | $1.16 \times 10^{+1}$ | $1.12 \times 10^{+1}$ | 2.56 | 9.90 | $1.38 \times 10^{+1}$ |
| | $A_6$ | $4.68 \times 10^{-2}$ | $-1.37 \times 10^{-1}$ | $-3.52 \times 10^{-1}$ | $5.55 \times 10^{-2}$ | $-2.70 \times 10^{-1}$ | $-6.62 \times 10^{-2}$ | $5.88 \times 10^{-2}$ | $-3.94 \times 10^{-2}$ | $-2.54 \times 10^{-1}$ |
| | $A_7$ | $2.11 \times 10^{-1}$ | $8.16 \times 10^{-1}$ | $7.84 \times 10^{-1}$ | $2.04 \times 10^{-1}$ | $8.70 \times 10^{-1}$ | $7.46 \times 10^{-1}$ | $2.01 \times 10^{-1}$ | $9.32 \times 10^{-1}$ | $7.42 \times 10^{-1}$ |
| | $A_8$ | $1.64 \times 10^{-1}$ | $2.63 \times 10^{-1}$ | $2.56 \times 10^{-1}$ | $1.68 \times 10^{-1}$ | $2.49 \times 10^{-1}$ | $2.56 \times 10^{-1}$ | $1.69 \times 10^{-1}$ | $2.35 \times 10^{-1}$ | $2.59 \times 10^{-1}$ |
| $\frac{(EA_{0.4})_r}{EA_{0.4}}$ | $A_1$ | $5.74 \times 10^{-1}$ | $-1.53 \times 10^{-1}$ | $-2.88 \times 10^{-1}$ | $5.82 \times 10^{-1}$ | $-3.85 \times 10^{-1}$ | $-3.70 \times 10^{-1}$ | $5.81 \times 10^{-1}$ | $-4.48 \times 10^{-1}$ | $-4.43 \times 10^{-1}$ |
| | $A_2$ | $1.37 \times 10^{-3}$ | $-1.58 \times 10^{-1}$ | $-3.34 \times 10^{-1}$ | $-5.76 \times 10^{-4}$ | $-1.49 \times 10^{-1}$ | $-3.58 \times 10^{-1}$ | $-1.15 \times 10^{-3}$ | $-1.38 \times 10^{-1}$ | $-3.08 \times 10^{-1}$ |
| | $A_3$ | $-5.86 \times 10^{-4}$ | $8.44 \times 10^{-4}$ | $6.42 \times 10^{-4}$ | $-6.26 \times 10^{-4}$ | $1.16 \times 10^{-3}$ | $5.47 \times 10^{-4}$ | $-5.61 \times 10^{-4}$ | $1.94 \times 10^{-3}$ | $2.56 \times 10^{-3}$ |
| | $A_4$ | $-1.88 \times 10^{-5}$ | $-4.47 \times 10^{-5}$ | $5.78 \times 10^{-5}$ | $-4.12 \times 10^{-5}$ | $1.41 \times 10^{-4}$ | $1.81 \times 10^{-4}$ | $-4.48 \times 10^{-5}$ | $1.61 \times 10^{-4}$ | $1.50 \times 10^{-4}$ |
| | $A_5$ | 2.19 | 1.29 | 2.18 | 2.08 | −1.56 | $6.68 \times 10^{-1}$ | 2.07 | −3.09 | −3.46 |
| | $A_6$ | $7.94 \times 10^{-2}$ | $4.52 \times 10^{-1}$ | $4.49 \times 10^{-1}$ | $1.25 \times 10^{-1}$ | $7.39 \times 10^{-1}$ | $6.88 \times 10^{-1}$ | $1.33 \times 10^{-1}$ | $8.25 \times 10^{-1}$ | 1.06 |
| | $A_7$ | $2.10 \times 10^{-1}$ | $9.25 \times 10^{-1}$ | $8.43 \times 10^{-1}$ | $1.89 \times 10^{-1}$ | 1.15 | $8.45 \times 10^{-1}$ | $1.83 \times 10^{-1}$ | 1.19 | $8.45 \times 10^{-1}$ |
| | $A_8$ | $1.69 \times 10^{-1}$ | $2.78 \times 10^{-1}$ | $2.88 \times 10^{-1}$ | $1.77 \times 10^{-1}$ | $2.86 \times 10^{-1}$ | $2.83 \times 10^{-1}$ | $1.79 \times 10^{-1}$ | $2.97 \times 10^{-1}$ | $2.65 \times 10^{-1}$ |
| $\frac{(EA_{0.8})_r}{EA_{0.8}}$ | $A_1$ | $5.96 \times 10^{-1}$ | $-2.33 \times 10^{-1}$ | $-3.10 \times 10^{-1}$ | $5.11 \times 10^{-1}$ | $-4.88 \times 10^{-1}$ | $-4.27 \times 10^{-1}$ | $4.57 \times 10^{-1}$ | $-4.96 \times 10^{-1}$ | $-4.32 \times 10^{-1}$ |
| | $A_2$ | $1.79 \times 10^{-2}$ | $-3.16 \times 10^{-1}$ | $-4.64 \times 10^{-1}$ | $-3.54 \times 10^{-2}$ | $-1.35 \times 10^{-1}$ | $-2.16 \times 10^{-1}$ | $-3.89 \times 10^{-2}$ | $-1.15 \times 10^{-1}$ | $-1.58 \times 10^{-1}$ |
| | $A_3$ | $-8.59 \times 10^{-4}$ | $1.51 \times 10^{-3}$ | $1.54 \times 10^{-3}$ | $-1.37 \times 10^{-3}$ | $3.87 \times 10^{-3}$ | $3.30 \times 10^{-3}$ | $-1.11 \times 10^{-3}$ | $4.18 \times 10^{-3}$ | $3.29 \times 10^{-3}$ |
| | $A_4$ | $2.76 \times 10^{-5}$ | $1.55 \times 10^{-4}$ | $1.06 \times 10^{-4}$ | $2.29 \times 10^{-4}$ | $2.43 \times 10^{-4}$ | $9.84 \times 10^{-5}$ | $3.04 \times 10^{-4}$ | $2.38 \times 10^{-4}$ | $8.56 \times 10^{-5}$ |
| | $A_5$ | 1.88 | −1.67 | $-9.57 \times 10^{-1}$ | 1.60 | −8.52 | −6.22 | 1.03 | −8.62 | −6.28 |
| | $A_6$ | $1.44 \times 10^{-1}$ | $5.98 \times 10^{-1}$ | $5.20 \times 10^{-1}$ | $2.43 \times 10^{-1}$ | $9.97 \times 10^{-1}$ | $7.77 \times 10^{-1}$ | $3.07 \times 10^{-1}$ | $9.58 \times 10^{-1}$ | $7.35 \times 10^{-1}$ |
| | $A_7$ | $2.03 \times 10^{-1}$ | $9.81 \times 10^{-1}$ | $8.88 \times 10^{-1}$ | $2.25 \times 10^{-1}$ | 1.09 | $8.52 \times 10^{-1}$ | $2.46 \times 10^{-1}$ | 1.08 | $8.43 \times 10^{-1}$ |
| | $A_8$ | $1.72 \times 10^{-1}$ | $3.18 \times 10^{-1}$ | $3.08 \times 10^{-1}$ | $1.95 \times 10^{-1}$ | $3.54 \times 10^{-1}$ | $3.11 \times 10^{-1}$ | $2.03 \times 10^{-1}$ | $3.52 \times 10^{-1}$ | $3.04 \times 10^{-1}$ |

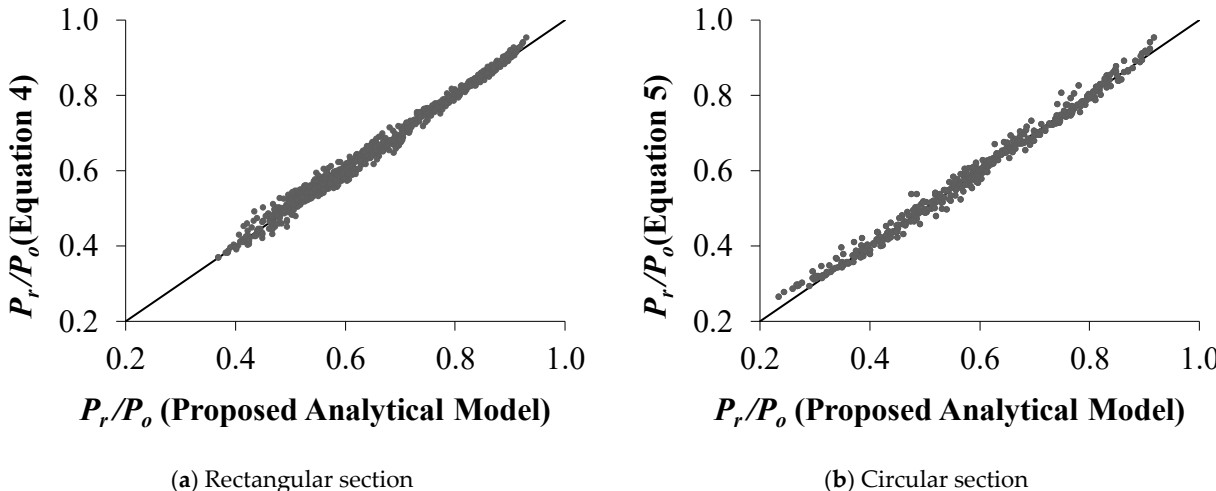

(**a**) Rectangular section         (**b**) Circular section

**Figure 15.** Validation of the proposed Equations (4) and (5) for residual capacity.

### 10.2. Circular Sections

Multiple linear regression analysis is also performed to propose a similar expression for the residual capacity and stiffness of the axially loaded circular RC columns, as shown in Equation (5).

$$\omega = B_1 + B_2\lambda + B_3 f'_c + B_4 f_y + B_5\rho + B_6\frac{\rho f_y}{f'_c} + B_7 D \tag{5}$$

where $D$ is the diameter of the cross-section (m). The coefficients ($B_{i\,=\,1,2,3,4,5,6,7}$) are given in Table 3 in a similar manner to the coefficients of the rectangular section. The line of equality plot reveals that the proposed expressions provide an excellent prediction of the capacity and stiffness compared to the results obtained from the analytical model, as illustrated in Figures 15b and 16b, respectively. The presence of outliers is almost negligible, which enhances the confidence of using the proposed expressions. The simplicity and robustness of the proposed expressions is an advantage for increasing their applicability during the design phase.

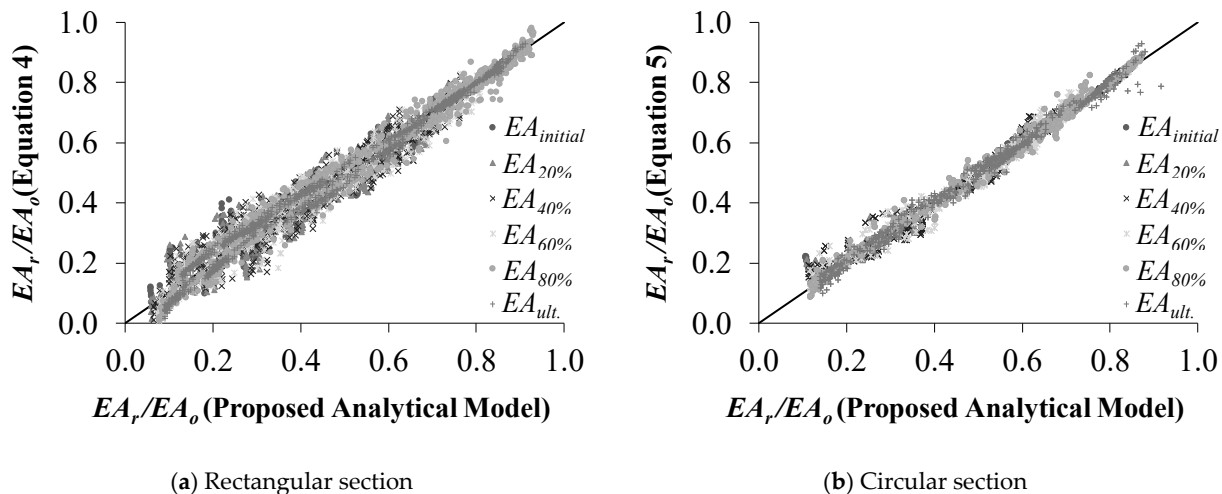

(a) Rectangular section      (b) Circular section

**Figure 16.** Validation of the proposed Equations (4) and (5) for residual axial stiffness.

**Table 3.** Coefficient of Equation (5) for circular sections.

| $\omega$ | $B_i$ | $R_D = 0$ (Unrestrained) | | | $R_D = 0.5$ (Partially Restrained) | | | $R_D = 1$ (Fully Restrained) | | |
|---|---|---|---|---|---|---|---|---|---|---|
| | | $t = 0.5$ hr | $t = 1.5$ hrs | $t = 2.5$ hrs | $t = 0.5$ hr | $t = 1.5$ hrs | $t = 2.5$ hrs | $t = 0.5$ hr | $t = 1.5$ hrs | $t = 2.5$ hrs |
| $\dfrac{P_r}{P_o}$ | $B_1$ | $6.70 \times 10^{-1}$ | $3.41 \times 10^{-1}$ | $1.96 \times 10^{-1}$ | $4.87 \times 10^{-1}$ | $1.74 \times 10^{-1}$ | $1.04 \times 10^{-1}$ | $4.41 \times 10^{-1}$ | $1.71 \times 10^{-1}$ | $4.63 \times 10^{-2}$ |
| | $B_2$ | $-2.52 \times 10^{-1}$ | $-3.97 \times 10^{-1}$ | $-3.96 \times 10^{-1}$ | $-1.44 \times 10^{-1}$ | $-1.44 \times 10^{-1}$ | $-1.43 \times 10^{-1}$ | $-1.17 \times 10^{-1}$ | $-1.24 \times 10^{-1}$ | $-8.38 \times 10^{-2}$ |
| | $B_3$ | $-1.76 \times 10^{-3}$ | $-1.91 \times 10^{-3}$ | $-1.86 \times 10^{-3}$ | $-1.77 \times 10^{-3}$ | $-2.32 \times 10^{-3}$ | $-2.40 \times 10^{-3}$ | $-1.92 \times 10^{-3}$ | $-2.62 \times 10^{-3}$ | $-7.54 \times 10^{-4}$ |
| | $B_4$ | $5.55 \times 10^{-6}$ | $9.80 \times 10^{-5}$ | $1.05 \times 10^{-4}$ | $1.14 \times 10^{-4}$ | $1.28 \times 10^{-4}$ | $1.49 \times 10^{-4}$ | $1.25 \times 10^{-4}$ | $7.86 \times 10^{-5}$ | $1.02 \times 10^{-4}$ |
| | $B_5$ | $5.75 \times 10^{-1}$ | $1.30$ | $1.43$ | $1.32$ | $1.60$ | $1.81$ | $1.54$ | $1.66$ | $8.87 \times 10^{-1}$ |
| | $B_6$ | $7.74 \times 10^{-2}$ | $8.00 \times 10^{-2}$ | $8.96 \times 10^{-2}$ | $7.71 \times 10^{-2}$ | $1.22 \times 10^{-1}$ | $1.06 \times 10^{-1}$ | $7.86 \times 10^{-2}$ | $1.37 \times 10^{-1}$ | $1.62 \times 10^{-1}$ |
| | $B_7$ | $3.88 \times 10^{-1}$ | $6.00 \times 10^{-1}$ | $6.53 \times 10^{-1}$ | $4.62 \times 10^{-1}$ | $6.24 \times 10^{-1}$ | $5.74 \times 10^{-1}$ | $4.95 \times 10^{-1}$ | $6.32 \times 10^{-1}$ | $6.12 \times 10^{-1}$ |
| $\dfrac{(EA_i)_r}{EA_i}$ | $B_1$ | $4.51 \times 10^{-1}$ | $7.79 \times 10^{-2}$ | $-1.64 \times 10^{-1}$ | $4.14 \times 10^{-1}$ | $7.25 \times 10^{-2}$ | $-1.59 \times 10^{-1}$ | $4.00 \times 10^{-1}$ | $7.57 \times 10^{-2}$ | $-1.59 \times 10^{-1}$ |
| | $B_2$ | $-5.91 \times 10^{-2}$ | $-4.15 \times 10^{-2}$ | $-6.81 \times 10^{-2}$ | $-3.41 \times 10^{-2}$ | $-1.79 \times 10^{-2}$ | $-4.96 \times 10^{-2}$ | $-2.83 \times 10^{-2}$ | $-1.56 \times 10^{-2}$ | $-5.38 \times 10^{-2}$ |
| | $B_3$ | $1.38 \times 10^{-4}$ | $-2.76 \times 10^{-4}$ | $-1.75 \times 10^{-3}$ | $6.76 \times 10^{-4}$ | $-2.02 \times 10^{-5}$ | $-1.79 \times 10^{-3}$ | $9.26 \times 10^{-4}$ | $-7.20 \times 10^{-6}$ | $-3.19 \times 10^{-3}$ |
| | $B_4$ | $-3.46 \times 10^{-5}$ | $-5.30 \times 10^{-5}$ | $1.60 \times 10^{-4}$ | $-4.79 \times 10^{-5}$ | $-7.07 \times 10^{-5}$ | $1.50 \times 10^{-4}$ | $-5.48 \times 10^{-5}$ | $-8.46 \times 10^{-5}$ | $2.40 \times 10^{-4}$ |
| | $B_5$ | $1.33$ | $2.41$ | $4.78$ | $1.40$ | $2.41$ | $4.78$ | $1.41$ | $2.42$ | $5.64$ |
| | $B_6$ | $3.36 \times 10^{-2}$ | $5.16 \times 10^{-2}$ | $-5.01 \times 10^{-2}$ | $5.10 \times 10^{-2}$ | $7.11 \times 10^{-2}$ | $-4.03 \times 10^{-2}$ | $5.80 \times 10^{-2}$ | $7.49 \times 10^{-2}$ | $-9.85 \times 10^{-2}$ |
| | $B_7$ | $4.98 \times 10^{-1}$ | $7.76 \times 10^{-1}$ | $9.21 \times 10^{-1}$ | $4.92 \times 10^{-1}$ | $7.42 \times 10^{-1}$ | $8.97 \times 10^{-1}$ | $4.94 \times 10^{-1}$ | $7.37 \times 10^{-1}$ | $9.04 \times 10^{-1}$ |
| $\dfrac{(EA_{0.4})_r}{EA_{0.4}}$ | $B_1$ | $4.95 \times 10^{-1}$ | $8.07 \times 10^{-2}$ | $-1.97 \times 10^{-1}$ | $4.42 \times 10^{-1}$ | $3.46 \times 10^{-2}$ | $-3.02 \times 10^{-1}$ | $4.23 \times 10^{-1}$ | $1.80 \times 10^{-3}$ | $-3.47 \times 10^{-1}$ |
| | $B_2$ | $-7.98 \times 10^{-2}$ | $-7.01 \times 10^{-2}$ | $-1.66 \times 10^{-1}$ | $-3.87 \times 10^{-2}$ | $-5.52 \times 10^{-2}$ | $-1.28 \times 10^{-1}$ | $-3.26 \times 10^{-2}$ | $-5.93 \times 10^{-2}$ | $-1.04 \times 10^{-1}$ |
| | $B_3$ | $4.74 \times 10^{-5}$ | $-1.17 \times 10^{-3}$ | $1.20 \times 10^{-3}$ | $8.94 \times 10^{-4}$ | $-1.66 \times 10^{-3}$ | $-3.48 \times 10^{-4}$ | $1.19 \times 10^{-3}$ | $-2.38 \times 10^{-3}$ | $2.62 \times 10^{-4}$ |
| | $B_4$ | $-3.24 \times 10^{-5}$ | $3.93 \times 10^{-5}$ | $-6.58 \times 10^{-5}$ | $-7.04 \times 10^{-5}$ | $1.35 \times 10^{-4}$ | $1.55 \times 10^{-4}$ | $-7.71 \times 10^{-5}$ | $2.33 \times 10^{-4}$ | $1.32 \times 10^{-4}$ |
| | $B_5$ | $1.31$ | $3.24$ | $1.31$ | $1.33$ | $4.32$ | $3.06$ | $1.37$ | $5.19$ | $2.26$ |
| | $B_6$ | $3.86 \times 10^{-2}$ | $1.76 \times 10^{-2}$ | $2.43 \times 10^{-1}$ | $6.98 \times 10^{-2}$ | $-1.77 \times 10^{-2}$ | $1.77 \times 10^{-1}$ | $7.61 \times 10^{-2}$ | $-6.54 \times 10^{-2}$ | $2.60 \times 10^{-1}$ |
| | $B_7$ | $4.69 \times 10^{-1}$ | $7.98 \times 10^{-1}$ | $1.03$ | $4.74 \times 10^{-1}$ | $7.77 \times 10^{-1}$ | $1.07$ | $4.78 \times 10^{-1}$ | $7.91 \times 10^{-1}$ | $1.11$ |
| $\dfrac{(EA_{0.8})_r}{EA_{0.8}}$ | $B_1$ | $5.44 \times 10^{-1}$ | $-1.01 \times 10^{-2}$ | $-2.63 \times 10^{-1}$ | $4.82 \times 10^{-1}$ | $-2.81 \times 10^{-1}$ | $-5.45 \times 10^{-1}$ | $4.53 \times 10^{-1}$ | $-3.12 \times 10^{-1}$ | $-5.66 \times 10^{-1}$ |
| | $B_2$ | $-8.70 \times 10^{-2}$ | $-2.37 \times 10^{-1}$ | $-3.61 \times 10^{-1}$ | $-5.11 \times 10^{-2}$ | $-1.22 \times 10^{-1}$ | $-1.19 \times 10^{-1}$ | $-4.65 \times 10^{-2}$ | $-9.72 \times 10^{-2}$ | $-8.29 \times 10^{-2}$ |
| | $B_3$ | $3.60 \times 10^{-5}$ | $9.20 \times 10^{-5}$ | $1.51 \times 10^{-3}$ | $3.15 \times 10^{-4}$ | $2.29 \times 10^{-3}$ | $3.44 \times 10^{-3}$ | $3.48 \times 10^{-4}$ | $2.34 \times 10^{-3}$ | $3.04 \times 10^{-3}$ |
| | $B_4$ | $-4.74 \times 10^{-5}$ | $1.39 \times 10^{-4}$ | $8.88 \times 10^{-5}$ | $-7.26 \times 10^{-5}$ | $2.60 \times 10^{-4}$ | $2.96 \times 10^{-4}$ | $-5.30 \times 10^{-5}$ | $2.62 \times 10^{-4}$ | $3.22 \times 10^{-4}$ |
| | $B_5$ | $1.22$ | $1.16$ | $-3.62 \times 10^{-1}$ | $1.55$ | $-4.88 \times 10^{-1}$ | $-2.10$ | $1.84$ | $-5.61 \times 10^{-1}$ | $-1.37$ |
| | $B_6$ | $6.96 \times 10^{-2}$ | $2.09 \times 10^{-1}$ | $3.45 \times 10^{-1}$ | $1.00 \times 10^{-1}$ | $3.91 \times 10^{-1}$ | $4.48 \times 10^{-1}$ | $9.48 \times 10^{-2}$ | $3.92 \times 10^{-1}$ | $3.57 \times 10^{-1}$ |
| | $B_7$ | $4.22 \times 10^{-1}$ | $9.14 \times 10^{-1}$ | $1.14$ | $4.47 \times 10^{-1}$ | $1.06$ | $1.25$ | $4.60 \times 10^{-1}$ | $1.09$ | $1.31$ |

## 11. Application of the Proposed Procedure

The proposed method is suitable to be implemented by engineers during the preliminary design phase for estimating the residual performance of RC frames exposed to

extreme standard fire conditions. The current study represents a step toward developing an integrated approach for considering all the components of the RC frames subjected to different loading conditions and exposed to various fire curves. This research assumes that the global behavior of the frame system is merely affected by the deterioration taking place in columns subjected to pure axial loads. This implies that beams and eccentrically loaded columns are either perfectly insulated against fire or are not exposed to critical temperatures capable of affecting their residual performance. The proposed procedure considers the interaction between the entire frame system and the fire-damaged columns in terms of connections' stiffness and load path. The fire-exposed columns are considered in the analysis as isolated members using an equivalent spring model whose stiffness is determined from the stiffness of the entire frame.

The steps required to adopt the proposed procedure are discussed in view of the 20-storey frame structure shown in Figure 17. The frame is composed of 8 m-long $300 \times 450$ mm RC beams made of normal-weight concrete with $f_c'$ of 35 MPa and reinforced with grade 400 MPa steel bars. The $300 \times 400$ mm columns are 3.6 m long with the reinforcement ratio of 0.04 and are constructed of the same materials as the beams. The moment of inertia of both member types is determined assuming cracked cross-sections (i.e., $I_{beam} = 0.35 I_g$ and $I_{column} = 0.7 I_g$), where $I_g$ is the gross moment of inertia of the considered member. The frame is loaded by subjecting the beams to a uniformly distributed load of 33 kN/m along the entire span. ASTM E119 standard fire is assumed to spread in the first floor of the building for 1.5 h, followed by a gradual cooling phase, according to ISO 834 specifications. Beams and corner columns are assumed to not be significantly influenced by fire, while the interior columns (i.e., columns IC1 and IC2) are exposed to fire from all sides. To determine the residual performance of the frame, the proposed procedure is discussed with reference to column IC1 in Figure 17. The structural analysis is performed using the commercially available ETABS [32] finite element software.

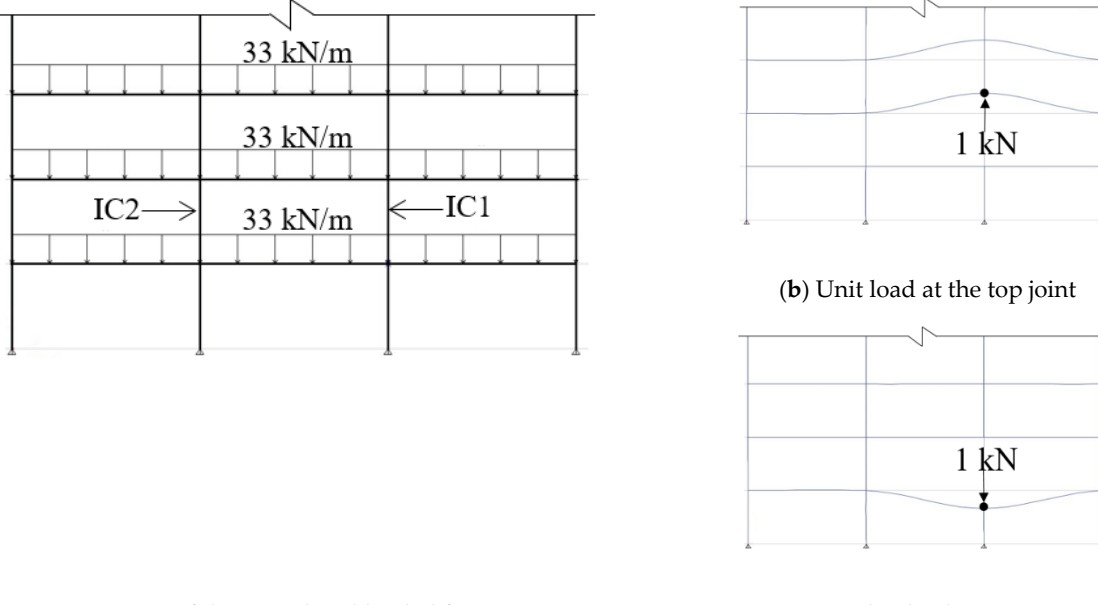

(**a**) Part of the considered loaded frame

(**b**) Unit load at the top joint

(**c**) Unit load at bottom joint

**Figure 17.** Description of the proposed analysis procedure.

(1) Determine the equivalent axial stiffness ($k_\delta$) of the spring shown in Figure 3b that represents the vertical stiffness of the structural system at that point. This is performed by replacing the examined column with a unit load acting at each joint individually, as shown in Figure 17b,c. The structural analysis is then performed on the frame to find the corresponding displacement of the considered joint. $k_\delta$ for each joint is calculated as the ratio between the unit load to the induced displacement. The total equivalent

axial stiffness ($k_\delta$) is then determined by considering the two joints as springs in series according to Equation (6).

$$k_\delta = \frac{(k_\delta)_1 (k_\delta)_2}{(k_\delta)_1 + (k_\delta)_2} \tag{6}$$

In this example, $(k_\delta)_1$ is determined as 10,000 kN/m, while $(k_\delta)_2$ is found to be 829,187 kN/m. Thus, $k_\delta$ for the isolated column model is 9881 kN/m.

(2) Calculate the axial restraint ration ($R_D$) as the ratio between $k_\delta$ calculated in step 1 and the axial stiffness of column per unit length ($EA/L$). In this example, $R_D$ is found to be 0.012.

(3) Determine the axial force acting on the considered column by performing structural analysis on the entire frame while the actual loads are added. Column IC1 in this example is subjected to an axial load of 2383 kN.

(4) Calculate the applied load level ($\lambda$) as the ratio between the applied load and the column axial capacity. In this example, $\lambda$ is determined as 0.4.

(5) Determine the residual axial capacity ($P_r$) and axial stiffness ($EA$)$_r$ of the considered column in view of the proposed expressions provided in Equation (4) along with Table 2 for rectangular sections. In this case, $\omega$ corresponding to the capacity and axial stiffness is 0.531 and 0.311, respectively. For columns IC1, this would be translated into a residual capacity and an axial stiffness of 3161 kN and 995,923,429 kN, respectively.

(6) Repeat the same procedure for all other axially loaded columns. In this example, the only other affected column is IC2.

(7) Adjust the axial capacity and stiffness of the considered columns in the structural program and repeat the analysis. Repeat steps 1 through 6 until the obtained variation in both capacity and stiffness for each column is within an acceptable tolerance.

(8) Once the residual behavior of all fire-damaged columns is adjusted in the program, the engineer can check the stresses, straining actions and deformation behavior of the frame in both the local and global levels.

## 12. Conclusions

In this paper, both thermal and sectional analyses are performed to determine the residual capacity and stiffness of fire-damaged rectangular and circular columns in typical RC frames. The temperature–load history experienced by the exposed members is considered in detail in the analytical study. The model is validated against relevant experimental studies found in the literature. A parametric study is carried out to determine the influence of various loading conditions and fire scenarios on the residual properties of the members. An objective-based method is then proposed to assist the engineers in evaluating the residual behavior of axially loaded RC columns considering an extreme standard fire scenario. The applicability of the proposed procedure is limited to RC columns made of normal-weight concrete and siliceous aggregate with fire durations up to 2.5 h, initial load level up to $0.4 f_c'$, section height between 400 mm and 800 mm, section width between 300 mm and 600 mm, section diameter between 350 mm and 650 mm, and steel reinforcement ratio between 2% and 4%. The main findings are as follows:

- The deterioration in both concrete and steel residual mechanical properties continues during the cooling phase as heat transfers not only to the atmosphere, but also to the colder inner elementary layers, raising their temperatures.
- Fire duration and cross-sectional dimensions are found to be the main parameters affecting the residual stiffness and capacity of the fire-exposed members.
- The influence of the initial load level on the residual stiffness and deformation behavior is noticeable as opposed to that on the residual capacity of the fire-exposed members.
- The permanent strength and stiffness reductions in the circular columns are found to be slightly higher than those with rectangular sections due to the higher maximum temperature reached within the circular sections.

**Author Contributions:** Conceptualization, M.A.Y.; data curation, M.M.A.A.; investigation, M.M.A.A.; writing—original draft preparation, M.M.A.A.; writing—review and editing, M.A.Y.; acquisition, M.A.Y. All authors have read and agreed to the published version of the manuscript.

**Funding:** This research was funded by the Natural Sciences and Engineering Research Council of Canada (NSERC), and Western University.

**Conflicts of Interest:** The authors declare that they have no known competing financial interest or personal relationships that could have appeared to influence the work reported in this paper.

### Nomenclature

| | |
|---|---|
| $A$ | area of the cross-section ($mm^2$) |
| $A_{sb}$ | area of bottom steel reinforcement ($mm^2$) |
| $A_{st}$ | area of top steel reinforcement ($mm^2$) |
| $b$ | width of rectangular cross-section (mm) |
| $D$ | diameter of circular cross-section (mm) |
| $E$ | modulus of elasticity of concrete (MPa) |
| $h$ | height of rectangular cross-section (mm) |
| $k_\delta$ | axial stiffness of the equivalent spring (N/mm) |
| $P_i$ | initial applied load (N) |
| $r$ | distance from the center of the circular cross-section to each layer (mm) |
| $R_D$ | axial restraint ratio |
| $t$ | time after the start of fire (hr) |
| $T_f$ | fire temperature (°C) |
| $T_o$ | room temperature (°C) |
| $\Delta\sigma$ | change in the applied load level |
| $\varepsilon_c$ | strain in concrete (mm/mm) |
| $\varepsilon_{cr}$ | creep strain (mm/mm) |
| $\varepsilon_{cuR}$ | residual ultimate strain of concrete (mm/mm) |
| $\varepsilon_{eq}$ | equivalent strain (mm/mm) |
| $\varepsilon_{LITS}$ | load-induced thermal strain (mm/mm) |
| $\varepsilon_m$ | mechanical strain (mm/mm) |
| $\varepsilon_R$ | total residual strain (mm/mm) |
| $\varepsilon_s$ | strain in steel bars (mm/mm) |
| $\varepsilon_t$ | total strain (mm/mm) |
| $\varepsilon_{th}$ | free thermal strain (mm/mm) |
| $\varepsilon_{thR}$ | residual free thermal strain (mm/mm) |
| $\varepsilon_{tr}$ | transient strain (mm/mm) |
| $\varepsilon_{trR}$ | residual transient strain (mm/mm) |
| $\varepsilon_\sigma$ | stress-related strain (mm/mm) |
| $\varepsilon_{\sigma i}$ | residual stress-induced strain (mm/mm) |
| $\lambda$ | axial load level |
| $\rho$ | steel reinforcement ratio |
| $\omega$ | residual capacity and axial stiffness ratio |

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
