# Peer review of "Residual Axial Behavior of Restrained Reinforced Concrete Columns Damaged by a Standard Fire"

_fire, doi:10.3390/fire5020042_

Round 1
Reviewer 1 Report
The paper is interesting and with in the scope of journal. The authors had made significant scientific contribution. However, some minor changes are suggested.
- Introduction is not sufficient and need improvements.
- Discuss significance of current study in the last paragraph of introduction.
- Figures quality are not good.
- Use color Figures so that it is easy for comparison.
- Table style should be corrected.
- Conclusion must be revised and should be written clearly in bullet point regarding different aspects. Write one bullet point for each aspect.
- What are future recommendations? Please add at the end of conclusions.
Reviewer 2 Report
The present manuscript deals with the residual axial behavior of concrete columns subjected to fire. The research is very interesting and relevant to the research and practitioners community. The authors have analyzed the cross-section under fire exposure by means of the boundary conditions method and have used an equivalent section for strength analysis. A very important contribution of the proposed research is the sensitivity analysis which was extended in the most involved parameters. Based on this analysis, the authors have proposed two parametric formulations for the residual capacity and strength of both circular and rectangular shape columns. This approach makes it easy to be implemented in simplified and fast estimation by practitioners.
Generally speaking, the text is well written and relevant authors' contribution is provided, thence in my opinion the research is publishable. The quality of the research and representation is high.
The drawbacks of the research are the following, and I do suggest to the authors to address these aspects.
- The authors have not considered spalling of the concrete, or in a simplified way, it would be not to consider the concrete cover during strength analysis. As your research is for axial load only, the concrete cover area could have a considerable portion.
- The sensitivity analysis is considered for different fire exposure times, but they have different exposure minutes under fire in other design codes. Does it mean in this case that interpolation of data will be applied? If yes, a linear correlation is suitable between the values of the parameter? This aspect should be highlighted.
- Different codes consider that a structural element could be under fire exposure not in all faces but in one two ets. How would this aspect affect your approach? Do you predict that the exposure of only one face could cause added moments and consequently buckling effects?
Reviewer 3 Report
The computational model proposed by the authors seems to be interesting and cognitively valuable, nevertheless, the idea of conducting calculations in this field in an iterative way, as a coupled mechanical-thermal analysis, has been known for a long time. It is good that the procedure described in the article has been properly validated based on the experimental results known to the authors. However, in each of the computational cases compared in this respect, the results obtained from the application of the analytical model proposed in this study gave a stiffness lower than that obtained experimentally, and this may prove to be dangerous for the user of the structure after the fire, considered in the analysis. A significant limitation of the proposed model is its reference to a very simple static scheme of the considered element, one that is associated with the action of only the axial force. It also does not consider the possible change in the flexural stiffness of the supports of the analyzed column with the temperature increase, because in the example a purely pinned support was assumed. The caption for Figure 2 seems too general. Is it really about interaction, or is it a diagram of individual iteration steps? The article under review is quite well written. The substantive reasoning seems to be legible and formally complete. Its possible publication, however, requires minor editorial corrections (such as, for example, changing the lowercase "p" in the description of the stress unit in Table 1 or incorrect marking of the diameter of rebar in line 404).
